# Ground-based millimetre-wave measurements of middle-atmospheric carbon monoxide above Ny-Ålesund (78.9° N, 11.9° E)

Niall J. Ryan[1], Mathias Palm[1], Christoph G. Hoffmann[2], Jens Goliasch[3], Justus Notholt[1]

[1]Institute of Environmental Physics, University of Bremen, Bremen, 28359, Germany
[2]Institute of Physics, University of Greifswald, Felix-Hausdorff-Str. 6, 17489, Greifswald, Germany
[3]RPG-Radiometer Physics GmbH, Werner-von-Siemens-Str. 4, 53340 Meckenheim, Germany

*Correspondence to*: Niall J. Ryan (n_ryan@iup.physik.uni-bremen.de)

**Abstract.** We present a new ground-based system for measurements of middle-atmospheric carbon monoxide (CO) at Ny-Ålesund, Svalbard, and the altitude profiles of CO volume mixing ratios (VMR) measured during the 2017/2018 winter.
The Carbon Monoxide Radiometer for Atmospheric Measurements (CORAM) records spectra from CO spectral emissions in the middle-atmosphere with the aid of a low-noise amplifier designed for the 230 GHz spectral region. Altitude profiles of CO VMRs are retrieved from the measured spectra using an optimal estimation inversion technique. The profiles in the current dataset have an average altitude range of 47-87 km, with special consideration to be given to data at > ~70 km altitude. The estimated uncertainty in the CO profile peaks at ~ 12 % of the a priori used in the inversion. The CORAM
profiles are compared to collocated CO measurements from the Microwave Limb Sounder (MLS) aboard the Aura satellite and show a difference of 7.4 – 16.1 %, with a maximum absolute difference of 2.5 ppmv at 86 km altitude. CO profiles are currently available at 1 hr resolution between November 2017 and January 2018. The instrument measures during Arctic winter because summer-time CO concentrations are so low as to be undetectable by CORAM.

## 1 Introduction

Millimetre-wave (also referred to as microwave) radiometers are powerful tools for measuring the composition of the atmosphere. This is particularly true for areas where there are prolonged night-time periods, such as the poles. The instruments can measure emissions from molecules in the atmosphere, in contrast to solar absorption measurements that rely on the sun. Coherent detection of the atmospheric signal, achieved through heterodyne receivers, and electronic manipulation of that signal, make it possible to detect and resolve spectral lines with very low intensities, especially when the electronics
are cooled to low temperatures, thus producing lower thermal noise (Janssen, 1993). Ground-based measurements in the thermal IR band generally do not have the capability to distinguish the mesospheric and stratospheric parts of the carbon monoxide (CO) profile (Kasai et al., 2005; Velazco et al., 2007).

Altitude profiles of CO concentrations in the middle atmosphere are useful in quantifying dynamical processes. Because the lifetime of CO during polar night is on the order of months (Solomon et al., 1985; Allen et al., 1999), it is a good tracer for
atmospheric dynamics. The generally increasing volume mixing ratio (VMR) of CO with altitude has a strong gradient,

which helps to identify the origin of increases or decreases in concentration. During polar night, CO concentrations increase in the middle atmosphere due to the vertical branch of the residual mean circulation bringing CO-rich air from higher altitudes (Smith et al., 2011; Garcia et al., 2014). Similarly, a decrease in middle-atmospheric CO in polar spring is linked to a change in direction of the residual mean circulation at this time. The breakup of the polar vortex in spring also allows more

CO-poor air to be transported poleward from lower latitudes (Manney et al., 2009; 2015), adding complexity to the quantitative link between dynamical processes and variations in CO. Changes in CO (and other tracers) VMRs can be caused by chemical production/loss (night-time CO is lost through reaction with a layer of hydroxyl at ~ 82 km (Solomon et al., 1985; Brinksma et al., 1998; Damiani et al., 2010; Ryan et al., 2018)), and by dynamical processes: vertical/horizontal advection, eddy transport, and to a lesser extent, molecular diffusion (Garcia and Solomon, 1983; Andrews et al., 1987;

Brasseur and Solomon, 2005; Smith et al., 2011). While vertical advection is, in general, the dominating process, modelling studies of middle-atmospheric CO indicate that the vertical transport rates calculated from trace gas measurements do not accurately represent the mean descent/ascent of the atmosphere because the 'true' effect of vertical advection is masked by other processes (Hoffmann, 2012; Ryan et al., 2018).

The general increase in middle-atmospheric CO VMR during polar night is seen in multiple datasets (e.g., Allen et al., 2000,

Forkman et al., 2005, Funke et al., 2009, Hoffmann et al., 2011, Ryan et al 2017), and the phenomenon has been observed for other tracers, e.g., $H_2O$ (Lee et al., 2011; Straub et al., 2012), $N_2O$, $CH_4$, and $H_2O$ (Nassar et al., 2005), and NO, $CH_4$, and $H_2O$ (Bailey et al., 2014). The calculated rates of vertical tracer transport in the above studies range from -1200 to +450 m/day (negative numbers indicate descent), with the values representing varying averages in space and/or time. Variations in tracer VMRs on smaller timescales (minutes to hours) can be caused by waves that displace air parcels from

their equilibrium positions and perturb trace gas profiles (e.g. Zhu and Holton, 1997; Ekermann et al., 1998; Fritts and Alexander, 2003; Noguchi et al., 2006; Chane Ming et al., 2016). Data from ground-based radiometers with high time resolution (order of an hour or less) have been used to investigate small periodic fluctuations in ozone ($O_3$) and water vapour (Hocke et al., 2006; Moreira et al., 2018; Schranz et al., 2018). The positive gradient of polar CO VMRs with altitude throughout the middle atmosphere, coupled with the time resolution of the presented measurement system at Ny-Ålesund

($\leq$ 1 hr), means that the dataset discussed here is well-suited to observing these periodic fluctuations, which are likely to be caused by vertical advection of air parcels by gravity waves (Zhu and Holton, 1997; Ekermann et al., 1998; Hocke et al., 2006). As with the ground-based and satellite-borne instruments in the works cited above, the analyses must be performed within the context of the limited spatial resolution of the measurements.

The Kiruna Microwave Radiometer, KIMRA, is also currently making measurements of middle-atmospheric CO at 67.8° N

(Raffalski et al, 2005; Hoffmann et al., 2011; Ryan et al., 2017), and the addition of a new instrument at Ny-Ålesund provides a needed increase in Arctic coverage and an excellent opportunity for comparison of CO at locations near the polar vortex edge and inside the vortex, particularly during dynamic events such as sudden stratospheric warmings. CO profiles from satellite measurements have been used regularly to study processes in the polar winter atmosphere (e.g. Damiani et al., 2014; Lee et al., 2011; Manney et al., 2009; McLandress et al., 2013), but recent ground-based CO datasets in the polar (and

nearby) regions have been sparse: The Onsala Space Observatory instrument (57° N, 12° E) (Forkman et al., 2012), which produced data for 2002 – 2008, and from 2014; The ground-based millimetre-wave spectrometer (GBMS) at Thule Air Base (76.5° N, 68.7° W), used to investigate the Arctic winter of 2001/2002 (Muscari et al., 2007) and the sudden stratospheric warming (SSW) in 2009 (Di Biagio et al., 2010); The British Antarctic Survey (BAS) radiometer data at Troll Station (72° S,

2.5° E) covers February 2008 to January 2010 (Straub et al., 2013). These instruments also measure the rotational transitions of CO and can operate during polar night.

The high time resolution of the CO Radiometer for Atmospheric Measurements (CORAM) is achieved primarily with a high-frequency low-noise amplifier (LNA), which operates on the atmospheric CO signal at 230.54 GHz before the signal is mixed with the radiometer's local oscillator. CORAM is discussed in Section 2, as well as the inversion method, CO profile

characteristics, and error estimates. Section 3 shows the results of a comparison with collocated data from the Microwave Limb Sounder (MLS). Section 4 shows the CORAM profile timeseries and discusses the usage of the data, and Section 5 offers some concluding remarks.

## 2 Instrument and measured data

### 2.1 CORAM

CORAM is a total-power radiometer housed at Ny-Ålesund, Svalbard (78.9° N, 11.9° E), and is part of the joint French-German Arctic Research Base, AWIPEV. CORAM measures the J = 2 → 1 rotational transition of CO at 230.54 GHz. The instrument was installed in 2017 and made first measurements of CO in the winter of 2017/2018. During the summer period, middle-atmospheric concentrations of CO are so small that they are not detectable by CORAM. The atmospheric signal enters the lab through a foam window that is transparent to millimetre-wave frequencies, and meets the pointing mirror of

CORAM, angled at 21° elevation. This angle was chosen by performing a series of atmospheric radiative transfer simulations at different elevation angles, using a climatological polar winter atmosphere, and determining which angle provided the strongest CO spectral line. The choice of angle is a trade-off of maximum path length through the target gas in the atmosphere, and minimum attenuation of the target signal by atmospheric water vapour that is primarily in the troposphere. The azimuth angle of the atmospheric signal is 113°, defined by the laboratory in which CORAM is held. After

the pointing mirror, the atmospheric signal is directed by a series of quasioptical components through a mylar window in a cryocooler and fed into a corrugated horn antenna. The quasioptical setup has an antenna pattern with a half-power-beam-width of ~ 5°. After the horn, the signal is amplified by a 230 GHz LNA. The unwanted sideband at ~ 213.5 GHz is supressed with a waveguide filter before the signal is mixed with the local oscillator (LO) signal (111 GHz) using a sub-harmonic mixer. Now at an intermediate frequency of 8.5 GHz, the signal exits the cooler and is amplified with another

LNA before being further downconverted to 0.5 GHz and analysed by a Fast Fourier Transform Spectrometer (FFTS).

Figure 1 shows a schematic drawing of the receiver including the components in the cryocooler, as well as a simplified version of the quasioptical layout. The alignment of the quasioptical components was checked using a laser positioned at the entrance to the cryocooler. The elevation angle of the instrument was measured using a self-levelling laser (Bosch GLL 3-80), which provides a horizontal line with an accuracy of 0.2 mm/m (0.2 mrad). Two horizontal lines, one directly from the laser and one passing through the quasioptical setup, were aligned on a screen approximately 5 m from the instrument. A sun scanning method has been used with other ground-based instruments for alignment and to identify a pointing offset, e.g., for MIAWARA-C (Straub et al. 2010) and GROMOS-C (Fernandez et al., 2015), for which the offsets in the elevation angle were found to be 0.01° and 0.07°, respectively.

The measured atmospheric signal is calibrated using two blackbody targets at known temperatures (measured with mounted sensors): a cold target in the cryocooler at ~ 70 K and a warm target at ~ 293 K. The integration time for each blackbody is the same as that for the atmospheric signal. A path length modulator is part of the setup that directs the atmospheric signal to the feedhorn, in order to reduce the amplitude of any standing waves in the quasioptics. The FFTS is an Acqiris AC240 and has a bandwidth of 1 GHz with 16384 channels, providing ~61 kHz resolution. A high spectral resolution, depending on the Doppler width of a spectral line (~300 kHz in this case), is required for resolving the mesospheric contribution to the spectrum. CORAM performs the Fourier Transform in real time and the fully resolved spectrum is stored. The cryocooler makes use of a CTI Cryogenics 350 CP coldhead and a CTI Cryogenics 8200 compressor, as well as a helium cooling machine.

Each electronic component in a signal chain will add noise to the atmospheric signal of interest, which will also be amplified with any subsequent amplifiers. Because of the better availability/price/quality of amplifiers that operated at several GHz, radiometers used for atmospheric measurements at frequencies > ~200 GHz have generally employed LNAs after the atmospheric signal has been mixed with the LO and has been downconverted to a lower frequency. The first LNA in CORAM, produced by Radiometer Physics GmbH (RPG), operates at a relatively high frequency of 230 GHz and allows for the atmospheric signal to be amplified before it encounters the mixer, ultimately providing an increased signal-to-noise ratio (SNR) for an atmospheric measurement. This configuration has been used before for similar instruments e.g., MIAWARA-C and GROMOS-C, which measure water vapour at 22 GHz, and ozone at 110 GHz, respectively.

An estimate of the improvement in the receiver temperature (Janssen, 1993) can be made using a noise temperature cascade analysis. A variation of Friis' equation (Vowinkel, 1988) for two components in succession is $T = T_1 + T_2/G_1$, where $T_1$, and $T_2$ are the respective noise temperatures of the first and second components, $G_1$ is the linear gain of the first component, and $T$ is the total noise temperature. The noise temperature of the LNA plus waveguide filter was measured to be 1350 K at room temperature, and the linear gain was measured at 158 (corresponding to 22 dB) (Fig. 2b). The noise temperature of the sub-harmonic mixer is ~ 1500 K at room temperature and has a linear gain of ~ 0.16 (corresponding to -8 dB). Applying Friis' equation with the LNA preceding the mixer gives a noise temperature of ~ 1360 K. The same calculation with the mixer as the first component gives a noise temperature of ~ 9800 K. The dominant contribution to the receiver temperature of CORAM is from the LNA/filter/mixer. Cooling the components can considerably reduce their noise temperature. Figure 2b

shows the noise temperature and gain of the LNA + filter, measured at room temperature. Figure 2c shows the receiver temperature for CORAM measured at the exit of the cryocooler, with the cryocooler components at a typical temperature of 39 K. At 8.5 GHz, the receiver temperature is below 350 K. Figure 2a shows the frequency response of the waveguide filter with a suppression of ~ -45 dB at 213.5 GHz.

The system temperature can be described as $T_{sys} = T_{rec} + T_a$ (Parrish et al., 1988; Janssen, 1993; Stanimirović et al., 2002). The receiver temperature, $T_{rec}$, considers the contributions from CORAM, and the antenna temperature, $T_a$, considers the contributions from the atmospheric background and signal being measured. The system temperature is related to the measurement time through the so-called ideal radiometer equation: $\sigma_T = T_{sys} / (Bt)^{1/2}$, where $\sigma_T$ is the statistical noise on a measured spectrum, $B$ is the frequency bandwidth of the measurement, and $t$ is the integration time for the measurement.

This relationship determines the measurement time required to provide a given SNR. The single sideband $T_{sys}$ for CORAM operating at Ny-Ålesund  is ~ 600 K. The atmospheric measurements are all made with the same elevation angle and so the individually recorded spectra can be averaged together to reduce the SNR. The measurements used here have been spectrally averaged over approximately 1 hour, including time used to calibrate the signal. Finer time resolutions that still yield usefully high SNRs are possible. Since $T_{sys}$, as defined here, contains a component from the atmospheric background, the

SNR of a given measurement will vary with the atmospheric conditions at the time, with a more opaque troposphere giving rise to a smaller SNR. An ad-hoc indication of "bad" weather conditions was found to be a measurement with a baseline temperature > 230 K, and these measurements were discarded.

## 2.2 CO profile retrieval

### 2.2.1 Defining the inversion problem

Schwarzchild's equation describes radiative transfer through a medium in local thermodynamic equilibrium. In the millimetre-wave region, at a given frequency, the measured intensity can be expressed in terms of brightness temperature, $T_b$, where

$$T_b = T_{b_0} e^{-\tau(l_0)} + \int_0^{l_0} T(l)\alpha(l)e^{-\tau(l)} dl, \tag{1}$$

with $l$ denoting the path through the atmosphere from a point $l_0$ to the measurement point at $l = 0$. The initial intensity is $T_{b_0}$,

the optical depth of the atmosphere is described by $\tau$, and the absorption coefficient is defined as $\alpha$. More details can be found in Janssen (1993) and references therein. $T_b$ in Eq. (1), as a function of frequency, is generally the mathematical description of the calibrated atmospheric spectrum, the antenna temperature ($T_a$) from Sect. 2.1. For a total power radiometer such as CORAM, the calibrated antenna temperature is found using:

$$T_a = \left(\frac{V_{atm} - V_c}{V_h - V_c}\right)(T_h - T_c) + T_c, \tag{2}$$

where $T_h$ and $T_c$ are the temperatures of the hot and cold calibration targets (Sect. 2.1), and $V_h$ and $V_c$ are the measured voltages when observing the hot and cold targets, respectively. $V_{atm}$ is the measured voltage when observing the atmosphere.

The desired quantity, the VMR of a trace gas, is contained within the description of the absorption coefficient, $\alpha$. Equation (1) must be inverted to retrieve this information. The form of Eq. (1) is that of a Fredholm integral of the second kind and is inherently sensitive to small perturbations (such as noise on a spectrum). To overcome this, the numerical inversion here is performed iteratively using a maximum a posteriori probability estimation.

**2.2.2 Inversion method**

Altitude profiles of CO VMR are retrieved from the measured spectra using an optimal estimation inversion technique (Rodgers, 2000). The method uses some a priori information of the state of the atmosphere to constrain the profile that is retrieved from the measured spectrum. The linear solution to the inversion problem can be expressed as $\hat{\mathbf{x}} = \mathbf{A}\mathbf{x} + (\mathbf{I} - \mathbf{A})\mathbf{x_a}$, where $\hat{\mathbf{x}}$ is the retrieved state vector (VMR profile), $\mathbf{x}$ is the true atmospheric state vector, $\mathbf{x_a}$ is the a

priori state vector, and $\mathbf{I}$ is the identity matrix. $\mathbf{A}$ is the averaging kernel matrix, which describes the sensitivity of a retrieved state to the true state (Rodgers, 2000). The sensitivity of the retrieved state at altitude $i$, to the true state at altitude $j$, is given by $\mathbf{A_{ij}} = \partial\hat{\mathbf{x}_i} / \partial\mathbf{x_j}$.

The inversions are performed with the Qpack2 package (Eriksson et al., 2005), which uses the Atmospheric Radiative Transfer Simulator (ARTS 2, Eriksson et al., 2011) to model the transfer of radiation through the earth's atmosphere. The a

priori CO profile used in the inversion is the average of one winter (September through April) of output from the Whole Atmosphere Community Climate Model (WACCM4) (Garcia et al., 2007), provided by Douglas Kinnison at the National Centre for atmospheric research (NCAR). Model output for the grid point encompassing Ny-Ålesund is used. The output is on a 132-layer pressure grid between approximately ground and 130 km altitude. A standard deviation of 100% at all altitudes was found to provide enough freedom for expected changes in CO VMR to be captured by the inversion, and to

give enough regularisation of the solution. Oscillations in the CO profile, a sign of over-fitting to the measurement (Rodgers, 2000), were found in several profiles. The oscillations were large in these cases so the CO profiles were considered unphysical and rejected. CO emissions are attenuated by absorption due to water vapour in the atmosphere (mostly in the troposphere) and this is accounted for by including the water vapour continuum by Rosenkranz (1998) in the forward model and inversion. $O_3$ is also simultaneously retrieved with CO, as an $O_3$ spectral line is centred at 231.28 GHz. The molecular

oxygen ($O_2$) and nitrogen ($N_2$) continua (Rosenkranz, 1993), as well as nitric acid ($HNO_3$) spectral lines, are included in the inversion but are not retrieved and are considered model parameters. The spectroscopic line data used here are from the high resolution transmission molecular absorption database (HITRAN) 2008 catalogue (Rothmann et al., 2009). The a priori information for $O_2$ and $O_3$, and water vapour is from the same WACCM4 run as for CO, and the information for $HNO_3$ and $N_2$ are from the FASCOD (Fast Atmospheric Signature Code) subarctic winter scenario (Anderson et al., 1986).

The information for the altitude, pressure, and temperature in an inversion is constructed from European Centre for Medium-Range Weather Forecasting (ECMWF) profiles and from the NRLMSISE-00 empirical model of the atmosphere (MSIS from herein) (Picone et al., 2002). ECMWF information is available daily at 6-hour intervals, beginning at midnight, and

covers up to 0.01 hPa altitude, and above that the temperature profile information is from MSIS. The temperature data are smoothed around the point where the profiles are merged to avoid discontinuities.

An estimate of the measurement noise on a spectrum is made by fitting a second-order polynomial to a wing of the spectrum and calculating the standard deviation of the fit. Qpack2 provides the capability to fit a series of functions to the baseline of

the measured spectra (a baseline fit) to account for errors in the baseline which are likely caused by standing waves in the instrument. The baseline fit is included in the optimal estimation and forms part of the overall fit to the measurement (inversion fit). All of the CORAM measured spectra were first inverted without a fit to the baseline and a periodogram of the residuals was evaluated to determine the periods of sinusoidal signatures in the baseline. Three primary sinusoids were found with respective estimated periods of 125, 62.5, and 41.67 MHz, and amplitudes of 0.2, 0.1, and 0.02 K. The periods of the

sinewaves are large compared with the width of the CO spectral line, which has a typical full-width at half-maximum (FWHM) of ~0.7 MHz, and so are uniquely distinguishable from it. The broad wings of a CO spectral line are produced by CO molecules at altitudes below the retrievable altitude limit of CORAM (approximately 47 km, see Sect. 2.3). A first order polynomial is also included in the baseline fit to account for offsets. The zeroth- and first-order coefficients have estimated uncertainties of 1 K and 0.5 K respectively.

The altitude grid for the forward model is between the ground and 125 km, with approximately equally-spaced points. The retrieval grid is between approximately 2 and 124 km, and is a 62-layer subset of the forward model grid. CO VMRs are retrieved as a fraction of their a priori for numerical stability due to the strong gradients in atmospheric CO. The inversion method is nonlinear and uses a Marquardt–Levenberg iterative minimisation scheme (Marquardt, 1963).

## 2.3 CO profile characteristics

The CORAM CO data spans November 18[th] 2017 to January 18[th] 2018. The instrument required maintenance after the latter date and was not in full operation for the remainder of the winter, unfortunately missing the SSW in February ☹. Nonetheless, the data shown here consists of 875 atmospheric profiles in that time, with time resolution of ~ 1 hr.

Figure 3 shows an example spectrum measured by CORAM on December 24[th] 2017, and the matching inversion fit and residual. The retrieved CO profile is also plotted in Fig. 3 alongside the a priori profile. The mean of the averaging kernels

for the whole CO data set are shown in Fig. 4 alongside the average of the estimated altitude resolution of the CO profiles. The estimated altitude resolution of the profiles is calculated here as the FWHM of the averaging kernels. A common way to estimate the altitude limits of a retrieved profile is to define the sum of the rows of the averaging kernels as the measurement response and assign a cut-off value. The measurement response can generally be thought of as a rough measure of the fraction of the retrieved state that comes from the data, rather than the a priori (Rodgers., 2000). As noted by Payne et al.

(2009), this is only a rough measure, and the measurement response often exceeds 1 at some altitudes. The choice of the cut-off value is rather arbitrary but 0.8 is regularly used (e.g., Forkman et al., 2012; Straub et al., 2013, Schranz et al., 2018), and is also used here. With the above definitions, the CO profiles from CORAM during winter 2017/2018 have an average altitude range of approximately 47 – 87 km, with an average altitude resolution varying between approximately 12.5 and

28 km over that range. The retrieval range can change depending on the distribution of CO in the atmosphere (the lower limit can decrease in altitude when there are higher CO values at lower altitudes) and the value provided here is the mean range over the time span of the data.

The retrieval limits will vary from measurement to measurement and individual profiles should be considered in combination with the accompanying averaging kernels (see Fig. 4). The centres of the averaging kernels, when represented in VMR, are shifted down in altitude compared to a representation in relative units (Hoffmann et al., 2011). The lower limit of the retrieval here is defined by the SNR in the measurement and the upper limit is set by a transition from a pressure broadening regime to a doppler broadening one. The result of this change is that, above approximately 70 km in the VMR representation, the centres of the averaging kernels do not increase in altitude with their corresponding retrieval altitudes. The retrieved CO values above ~ 70 km altitude do contain information from the atmosphere that corresponds with the retrieval altitude, but the VMR representation of the profile should be considered with care. Hoffmann et al. (2011) provides a detailed discussion on the representation of data for ground-based CO measurements. Hoffmann emphasises that the limited vertical resolution of the data must be taken into account for the use and interpretation of the data by considering each realisation of the averaging kernels, and so the a priori and averaging kernels form an essential part of the dataset.

## 2.4 CO profile error estimates

The error contributions to the CO profiles are calculated using OEM error definitions, which are defined in detail in Rodgers (2000). The estimates of the errors are found by perturbing the inputs to the inversion, using the following uncertainties. Error in the temperature profile is the same as that used in Hoffmann (2011): 10% above 100 km, 5% below 80 km, and linearly interpolated in between. An uncertainty of 1° is chosen for the pointing of the instrument to the sky, an overestimate of the motor (Faulhaber 3564K024B CS) uncertainty by an order of magnitude, to account for changes that may occur in the orientation of the instrument table. The uncertainty in the warm and cold calibration targets is 2 K, an overestimate that accounts for variations and drifts in the temperatures. The HITRAN 2008 catalogue is used for uncertainties in the CO line parameters: 1% for the line intensity, 2% for the air broadening parameter and 5% for the temperature dependence of the air broadening. The uncertainties related to self-broadening of CO are not considered due to the relatively low concentration of the gas (Ryan and Walker, 2015). The uncertainty in the line position is ignored because the frequency grid used in the inversion is shifted to centre a measurement.

The error estimates, including the average of the error arising from statistical noise on the spectrum, are plotted in Fig. 5. The sum in quadrature of the error estimates is also plotted, as well as the a priori CO profile for the data set. The statistical noise on the spectrum and the uncertainty in the temperature profile are the biggest contributors to the total error profile, with the temperature error surpassing that of the spectrum noise at ~ 84 km, near the average upper retrieval altitude limit. As a fraction of the a priori profile, the total error estimate has a maximum at ~ 12% at ~ 48 km, near the average lower retrieval altitude limit, and there is also a peak of 11.5% near 70 km altitude. The uncertainty in the temperature profile begins to become more pronounced above 50 km altitude.

## 3 Comparison with Aura MLS

MLS is a radiometer aboard the Aura satellite. A description of the instrument is given in Waters et al. (2006). Version 4.2 of the MLS CO data (Schwartz et al., 2015) is used here and is described in Livesey et al. (2015). The atmospheric pressure range of the data is 215 - 0.0046 hPa. The precision of the CO VMR profile reaches a maximum (largest) value of 1.1 ppmv
at the upper limit of the MLS CO retrieval altitude. The data have an estimated average positive bias (larger VMRs) of ~17% above 40 km altitude (Sheese et al., 2017), ~~positive bias of 20% in the middle atmosphere (larger VMRs),~~ compared to the Atmospheric Chemistry Experiment – Fourier Transform Spectrometer (ACE-FTS) satellite instrument ~~(Livesey et al., 2015)~~. ~~This~~ Sheese et al. (2017) use Version 3.3/3.4 MLS CO data, which shows good agreement with Version 4.2 (Livesey et al., 2018), and have not included data from the summer months when CO concentrations are very low. ~~bias is estimated~~
~~from a study of Version 2.2 of MLS CO data (Pumphrey et al., 2007), which showed a positive bias of 30 %. Subsequent versions of MLS CO, including the version used here, show a slight decrease in the CO VMR, bringing the values closer to those of ACE-FTS.~~

## 3.1 Colocated measurement comparison

MLS measurements are subset to within ±2° latitude and ±10° longitude of CORAM, calculated at 60 km altitude along the
line of sight of CORAM (~ 156 km horizontally from the lab). The CO VMRs are expected to vary more in latitude than in longitude because the atmospheric composition generally varies more in the meridional direction compared to the zonal. A longitude space of ±5° was tested but there were not significant changes to the results shown here and the number of coincident MLS measurements were halved. Above 0.001 hPa, MLS CO profiles have a constant VMR value. Because CORAM has some sensitivity to CO at these altitudes, the MLS profiles were instead linearly extrapolated in pressure space
above 0.001 hPa. A more physically realistic profile shape is produced, an example of which can be seen in Fig. 4 of Ryan et al. (2017). To reduce the effect of atmospheric variability between individual measurement locations, the CORAM and MLS profiles are averaged by day to produce daily mean profiles. These MLS profiles were smoothed (Rodgers, 2000) with the averaging kernels of the corresponding CORAM profiles to account for the finer altitude resolution of MLS CO profiles: 6-7 km in the upper mesosphere and 3.5 to 5 km in the upper troposphere to the lower mesosphere (Livesey et al., 201~~5~~8).
Figure 6 shows the mean CO profiles for CORAM and MLS over the time of measurement overlap (November 19[th] to January 18[th]), as well as the absolute and percentage (relative to the mean of the MLS and CORAM profiles) differences in the profiles. The correlation of the CO VMRs at each retrieval altitude is also plotted. The maximum absolute difference in the mean profiles is 2.5 ppmv at 86 km altitude, corresponding to an 11.3% difference. The percentage difference varies between ~ 7.4 % at the lowest retrieval altitude and 16.1% at 72 km, with MLS having a low bias in comparison to CORAM
over the entire altitude range. This contrasts with the estimated high bias of MLS compared to ACE-FTS, mentioned above. The standard deviation of the differences in the profiles is largest (in percentage) at 58 km with a value of 14.4 %. The correlation of the CORAM and smoothed MLS CO profiles is greater than 0.80 at all retrieval altitudes, reaching a

maximum of 0.92 at 47 km. After smoothing, the MLS and CORAM data are not truly independent, so the correlation of CORAM with the unsmoothed MLS data is also calculated and shows more variation over the retrievable altitude range, with a minimum of 0.59 and a maximum of 0.81. The statistics here show some similarities to the comparison of MLS CO and ground-based CO measurements from KIMRA (67.8° N), where MLS showed a low bias (peaking at ~ 0.65 ppmv) up to

~ 74 km, with a maximum relative bias of 22% at 60 km (Ryan et al., 2017). The correlation between KIMRA and MLS was slightly higher than that for CORAM and MLS, remaining greater than 0.90 up to 82 km altitude.

Figure 7 shows the daily time series of the MLS and CORAM profiles at 48, 58, 68, 78, and 88 km. The largest differences in CO are found at higher altitudes ($\geq$ 68 km) in November and the first days of December, after which the values become closer in VMR, indicating better agreement between the instruments. The reason for the larger difference over this time is

unknown, but it is clear that these high values contribute to the bias between the instruments shown in Fig. 6. Despite the absolute differences, a similar variability in CO is captured by both instruments over the whole time series.

## 4 CORAM data and usage

Figure 8 shows the currently available CORAM CO data for winter 2017/2018 at 1 hr time resolution. The anomalously high values above ~ 70 km altitude are visible in November and first days of December. At lower altitudes over this time, there is

still some downwelling of CO due to the residual mean circulation, before a levelling off in mid-December. Figure 8 also shows a 43-hour segment of the data beginning at 5 pm on December 31[st] 2017, to illustrate the advantage of continuous measurements. Below ~75 km altitude, there is apparent downwelling of CO for about the first 25 hours, peaking before VMR values start to decrease over the next 18 hours. There are two relatively strong increases in lower-altitude CO at approximately 2pm January 1[st] and 1am January 2[nd], evident from the 2.3 ppmv contour line moving down from 60 to 50 km

altitude. Over this same time, between 60 and 70 km, there is an oscillation in the 4.1 and 6 ppmv contour lines, with peaks occurring every 1-2 hours. The VMR values above approximately 75 km tend to show similar short-timescale variations but with opposite sign, i.e., a peak at a higher altitude corresponds with a trough at a lower altitude. This inverted pattern is observable over the whole 43-hour time period. Variations on these timescales cannot be directly observed by non-geostationary satellites, illustrating the unique capability of ground-based instruments.

These are broad descriptions of the data because one cannot fully characterise the variations in CO without the use of other data sources and model output. Variations on the timescales of an hour to weeks are visible in the data and require detailed study to elucidate the underlying dynamical processes, such as polar vortex shift, Rossby wave activity, SSW events, gravity wave perturbations (time scales of minutes to hours). Peridocities in trace gas data have previously been analysed using spectral decomposition techniques on ground-based measurements of water vapour and ozone (e.g., Hocke et al., 2006,

Struder et al., 2014, Schranz et al., 2019) to identify waves with periods of days to weeks.

As mentioned in Sect. 2.3, the CORAM profiles should be used with consideration of the accompanying averaging kernels. Ground-based measurements have limited altitude resolution, often much coarser than the altitude grids onto which the data

is retrieved. The representation of the data on a fine grid adds stability to the inversion (Eriksson, 1999) and can give rise to substantial smoothing error in the profiles (Rodgers, 2000). The smoothing error can be accounted for when comparing CORAM to instruments with higher resolution by convolving the data from the other instrument with the CORAM averaging kernels, as was done for MLS in Sect. 3. The error should be assessed if one is to use the CO profiles without considering the sensitivity distribution described by the averaging kernels. This is not a recommended use of the data and why the smoothing error is not assessed in Sect. 2.4. In other words, if one is to say something of a CORAM CO VMR at a given grid point, on must be aware that the VMR value at that grid point contains information from a range of altitudes, with a sensitivity governed by the associated averaging kernel.

CORAM profiles can be used independently to describe changes in CO over time, providing the averaging kernels do not significantly change over this time, which would change the measurement response. The measurement response for CORAM should not show significantly variation inside the retrievable altitude range but care should be taken at altitudes near the edges of the retrieval range of the profiles, where the measurement response has a strong gradient and can change quickly when there are rapid changes in CO concentrations at those altitudes. CORAM is currently under maintenance due to a fault in the LO signal generator and is expected to be back in operation for the winter of 2019/2020 and beyond.

## 5 Conclusion and future work

This work presents a new ground-based radiometer, CORAM, that has been installed at the high-Arctic location of Ny-Ålesund, 78.9° N, for the measurement of middle-atmospheric CO. The instrument makes use of a high-frequency LNA, before the downconversion of the atmospheric signal, to achieve high SNRs at time resolutions on the order of an hour or less. CO profiles were retrieved from measurements in the Arctic winter of 2017/2018. Error estimates show that the uncertainty in the temperature input for the inversions and the statistical noise on the spectrum are the largest contributions to the error budget, giving a maximum in the error profile of ~ 12 % of the a priori profile. The mean of the averaging kernel matrix for the CORAM dataset gives an average retrieval altitude range of 47-87 km with an average altitude resolution of 12.5 to 28 km over this range. Data at higher altitudes should be treated with care as the VMR representation of the averaging kernels do not peak at the corresponding retrieval grid points above ~70 km altitude. A comparison with MLS shows a negative bias (MLS - CORAM) at all altitudes, with a maximum of 16.1 % of the average profiles occurring at 72 km altitude. A comparison of the instruments' time series indicate abnormally high CO measured by CORAM above ~ 68 km in November 2017 that contributes to the observed bias, after which the MLS and CORAM values show improved agreement. Correlations between the instruments range from 0.80 to 0.92 over CORAMs retrievable altitude range for MLS data smoothed with the CORAM averaging kernels, and from 0.59 to 0.81 when using the unsmoothed MLS data. CO profiles above Ny-Ålesund with a 1 hr time resolution between November 19[th] 2017 and January 18[th] 2018 are currently available. Future work with CORAM will include: Integration of a newly manufactured local oscillator due to a failure of the original, and investigation of possible attenuation of the atmospheric signal by the laboratory foam window.

## Data availability

CORAM Level 2 data, including averaging kernels and metadata, are available on request via N. Ryan (n_ryan@iup.physik.uni-bremen.de) and M. Palm (mathias@iup.physik.uni-bremen.de). A public data archive is planned for after CORAM resumes operation in the winter of 2019/2020. The Aura MLS v4.2 data are available from the Goddard Earth

Sciences Data and Information Center at https://disc.gsfc.nasa.gov.

## Author contribution

M. Palm and C. Hoffmann designed the project. C. Hoffmann, N. Ryan, and J. Goliasch designed and built CORAM. N. Ryan developed the inversion setups for CORAM and performed the comparisons. N. Ryan installed CORAM at Ny-Ålesund, and the instrument was maintained by M. Palm. J. Notholt provided valuable feedback on the project. N. Ryan

prepared the manuscript with contributions from co-authors.

## Acknowledgements

This work has been funded by the German Federal Ministry of Education and Research (BMBF) through the research project: Role Of the Middle atmosphere in Climate (ROMIC), sub-project: ROMICCO, project number: 01LG1214A, as well as by a grant from the Canadian Space Agency. We would like to express our gratitude to the MLS teams for making

their CO product available. We would also like to thank the ECMWF and MSIS teams for making their products available, as well as the Qpack and ARTS communities for making their software available. We thank the AWIPEV staff for all of the help provided at Ny-Ålesund, particularly Benoit Laurent, who aided in the installation and maintenance of CORAM at Ny-Ålesund. A special thank you to Eloise and Elmo Ryan, for all of their support.

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

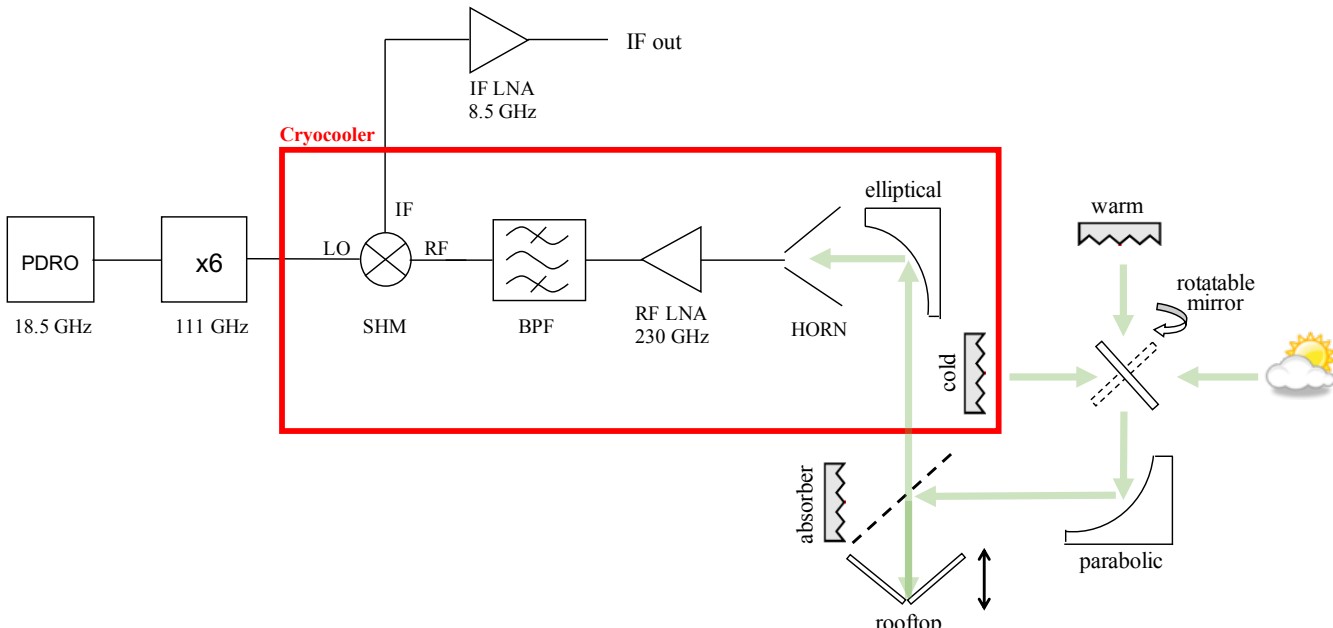

**Figure 1: Schematic of the CORAM receiver and simplified version of the quasioptics. A rotatable mirror selects a signal from either the atmosphere, warm target, or cold target. The signal is directed by a parabolic mirror to a path length modulator that comprises a polarising wire grid, an absorber, and an oscillating rooftop mirror. The signal passes through a window in the cryocooler where it is directed to the receiver with an elliptical mirror. The signal enters the corrugated feed horn and encounters the RF LNA, a waveguide filter (BPF), and a sub-harmonic mixer (SHM). At the SHM the signal is downconverted to an intermediate frequency (IF) of 8.5 GHz. The IF signal exits the cryocooler and passes through a room temperature LNA. The RF (atmospheric) signal is mixed at the SHM with a local oscillator (LO), which is an 18.5 GHz signal from a phase-locked dielectric resonator oscillator (PDRO) that is passed through a x6 frequency multiplier, to provide 111 GHz. The IF out signal will be further downconverted to 0.5 GHz before being analysed by the Fast Fourier Transform Spectrometer (not shown here). Further details on quasioptical components can be found in Goldsmith (1998).**

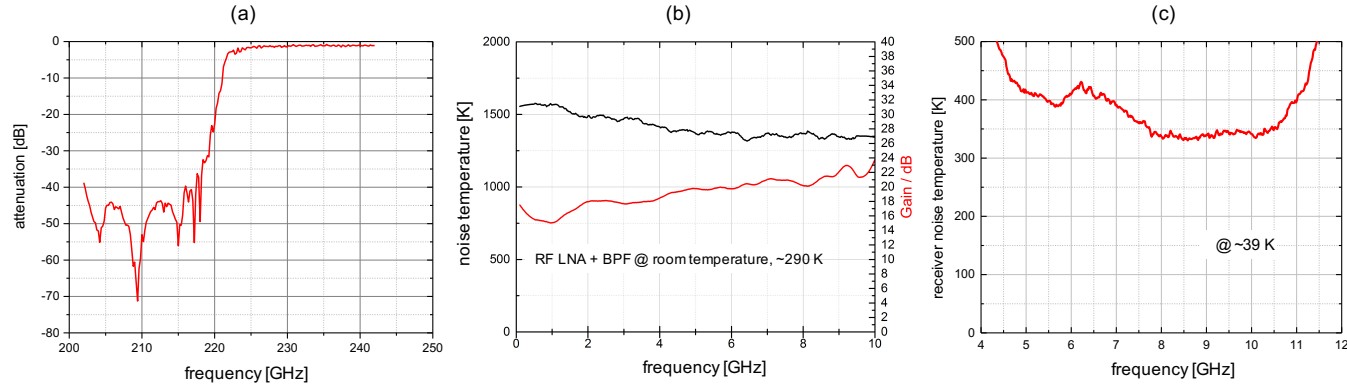

**Figure 2: (a) The frequency response of the waveguide filter (BPF in Fig. 1) used in CORAM to supress the unwanted sideband signal at 213.5 GHz. (b) The noise temperature and gain of the RF LNA + BPF (Fig. 1) at room temperature. (c) The noise temperature for CORAM after downconversion to 8.5 GHz. This measurement is made after the first IF LNA (Fig. 1) and before the second downconversion to 0.5 GHz. The cryocooler components are at 39 K. The single sideband system temperature for CORAM is ~ 600 K (Sect. 2.1).**

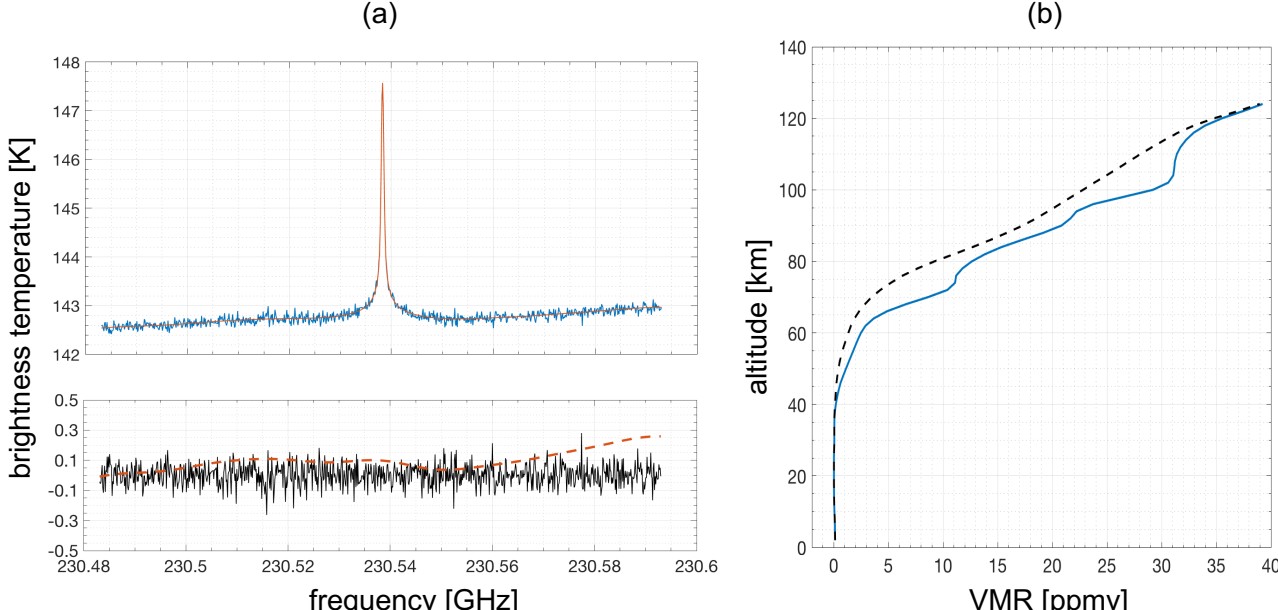

**Figure 3: (a)** Upper: an example spectrum measured by CORAM on Dec 24[th] 2017 between 20:04 and 21:03 UTC. The inversion fit to the measurement is shown (smoother red line). Lower: the residual of the measurement and the inversion fit (solid black line). The dashed red line shows the baseline fit for the inversion, which is part of the inversion fit shown in the upper panel (Sect. 2.2.2). **(b)** The CO profile retrieved from the measurement (solid blue) and the a priori profile that is used as input to the inversion (dashed black).

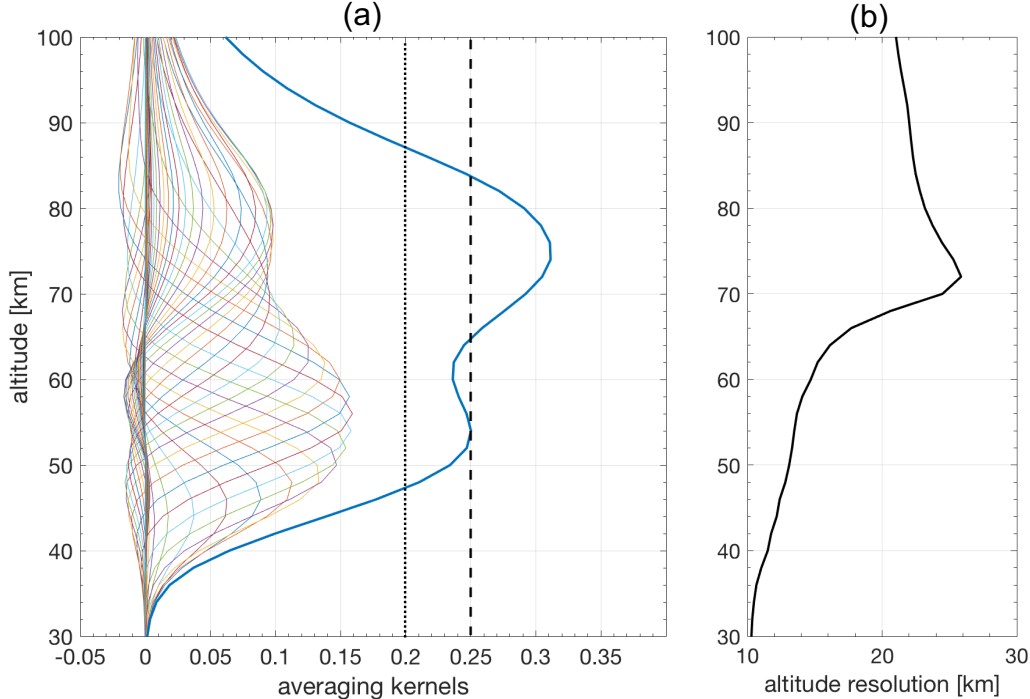

**Figure 4: The mean averaging kernels for the CORAM inversions. The measurement response (sum of the rows of the averaging kernels) divided by 4 is shown in thick solid blue. The dashed black line and the dotted black line indicate a measurement response of 1.0 and 0.8, respectively. (b) The mean altitude resolution of the CORAM CO profiles, calculated from the FWHM of the averaging kernels.**

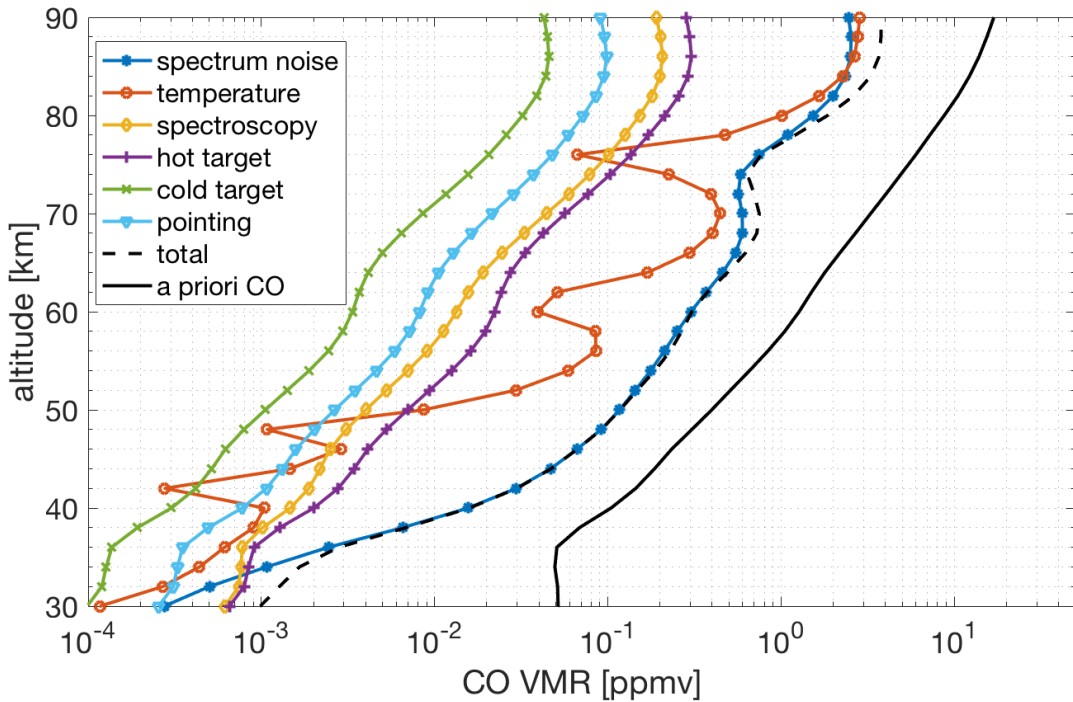

**Figure 5: The estimated error contributions to the CORAM CO profiles. The spectrum noise is calculated as an average of the noise on all CORAM measurements, and the other estimates are calculated through perturbations about the a priori CO profile (Sect. 2.4).**

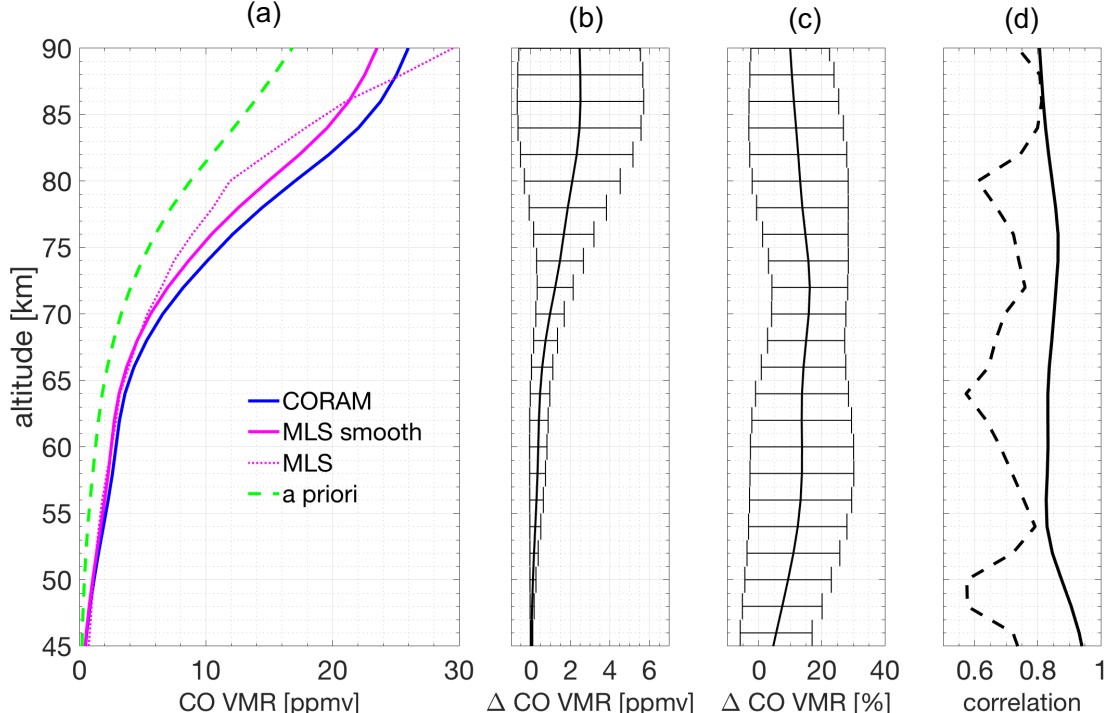

**Figure 6: (a) The mean of the daily CORAM and MLS CO profiles above Ny-Ålesund. The mean of the unsmoothed MLS profiles is also shown as well as the a prioiri profile used for the CORAM inversions. (b) The absolute difference of the mean CORAM and smoothed MLS profiles, with the standard deviation of the differences as the whiskers on the line. (c) The same as for (b) but with the difference as a percentage of the mean CORAM and MLS profiles. (d) The correlation coefficients of the CORAM and smoothed MLS data (solid) and unsmoothed MLS data (dashed).**

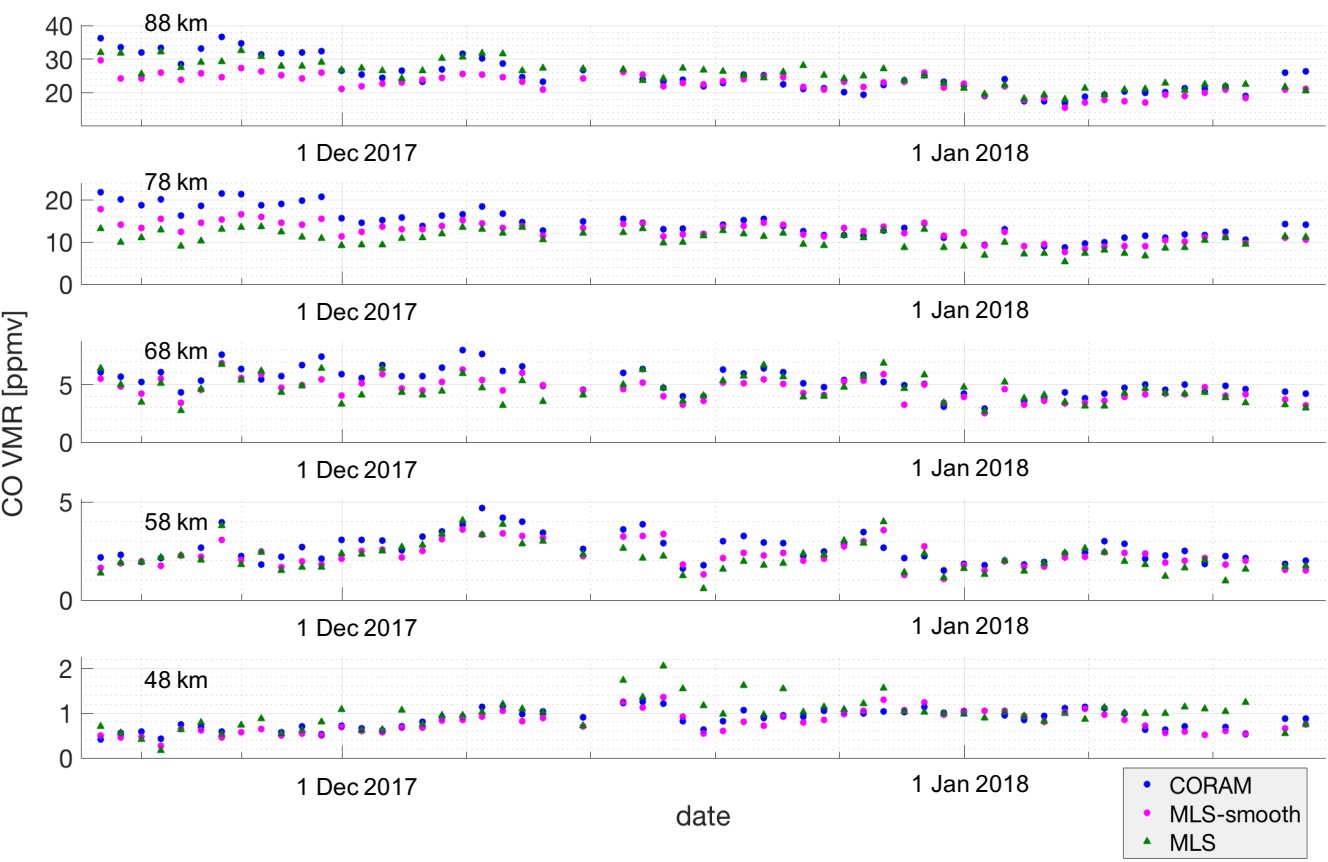

**Figure 7: Time series of the daily CORAM and MLS CO VMR values at altitudes of 48, 58, 68, 78, and 88 km.**

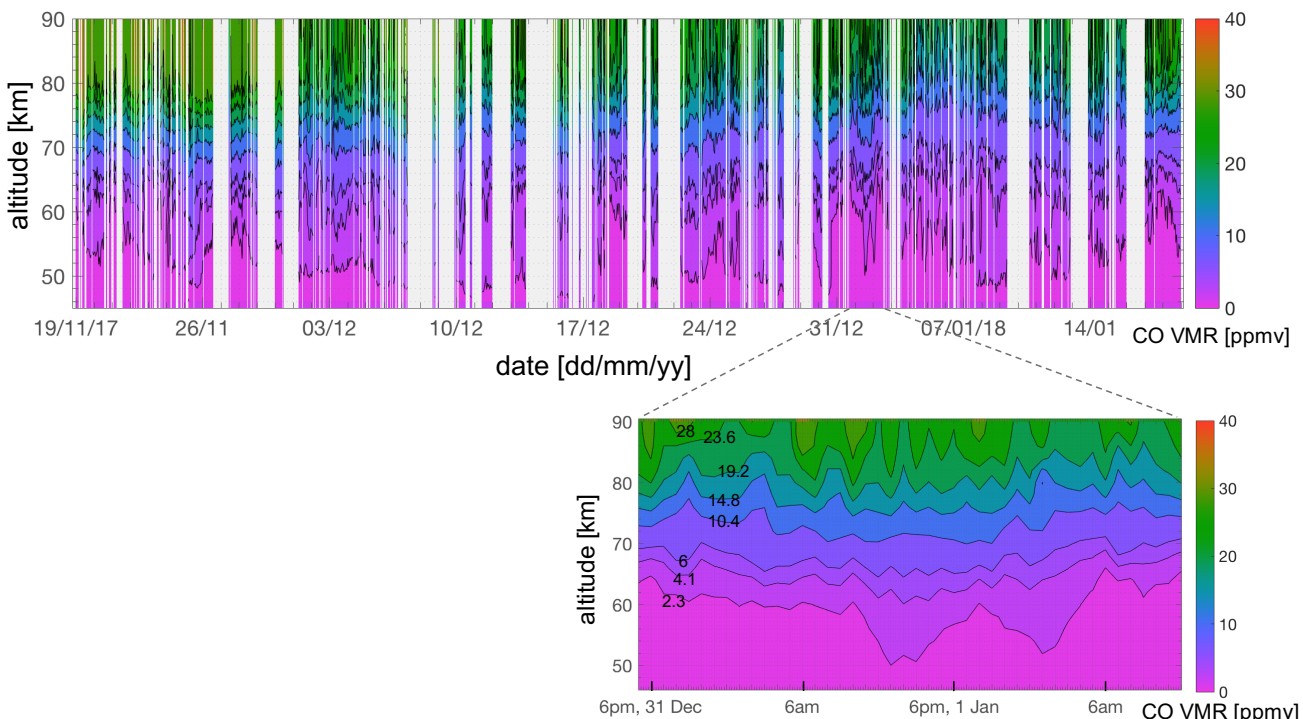

**Figure 8: CORAM CO profiles at 1 hr resolution from mid-November 2017 to mid-January 2018. Blank areas are gaps in the data record. The zoomed-in plot shows measurements over a 42-hour period beginning at 6pm on December 31st 2017. The Isoluminant colour map from Kindlmann et al. (2002) is used. Contour values are [0.4, 2.3, 4.1, 6.0, 10.4, 14.8, 19.2, 23.6, 28.0], chosen and filled for readability. Gaps in the data record correspond to periods of non-operation or bad measurement data.**