# Peer review of "Ground-based millimetre-wave measurements of middle-atmospheric carbon monoxide above Ny-Ålesund (78.9° N, 11.9° E)"

_Atmospheric Measurement Techniques, 2019_

## Referee Comment (RC1) · Anonymous Referee #1 · 11 Apr 2019

**Journal:** Atmos. Meas. Tech. Discuss.
**Title:** Ground-based measurements of middle-atmospheric carbon monoxide above Ny-Ålesund (78.9° N, 11.9° E)
**Author(s):** Niall J. Ryan, Mathias Palm, Christoph G. Hoffmann, Jens Goliasch, and Justus Notholt
**MS No.:** amt-2019-121 (https://doi.org/10.5194/amt-2019-121)
**MS Type:** Research article

**Review**

**General Comments**

The paper presents carbon monoxide (CO) measurements in the polar upper stratosphere and mesosphere made using a new ground-based millimetre-wave radiometer (CORAM). Atmospheric observations recorded during the 2017–18 winter from Ny-Ålesund are analysed using optimal estimation retrieval techniques to determine vertical profiles of CO volume mixing ratio. The precision of the measurements is estimated and CORAM profiles are compared with overlapping CO measurements by the Aura MLS satellite instrument.

Measurements of CO in the polar middle atmosphere are important as the gas is sufficiently long-lived to be used as a tracer for characterising dynamical and transport processes associated with the winter-time polar vortex, and atmospheric wave and tide activity. The structure and extent of the polar vortex above Antarctica, and its more dynamic counterpart in the Northern hemisphere affect global circulation patterns, stratospheric ozone abundances, and atmospheric heating rates. Targeted ground-based observations of CO, such as those presented here, complement the much larger geographical coverage of satellite remote sensing datasets. The ground-based instruments provide continuous observations with the potential to resolve features occurring on short timescales.

Overall the paper is reasonably well written and presented, with adequate description of the observations, data-sets, discussion of the results, and citing of prior work. The paper covers the development, inter-comparison, and validation of an atmospheric measurement system and this fits well with the subject areas of the journal. My main criticism is the lack of important details about the CORAM instrument itself. The main novelty in the work is the new design of this radiometer that incorporates optical/electronic components. The authors suggest that this design improves the performance/cost over previous heterodyne radiometers used for this type of measurement. As indicated in my specific comments below, the sections (2.1 and Figure 1) of the paper describing the new instrument need expanding with further technical details. The manuscript would benefit from a more thorough description of existing CO measurement systems including those utilising the thermal IR bands of CO to make middle atmosphere observations. The conclusions section should include clear statements as to how well the anticipated improvements in performance were achieved. I've also identified a number of areas in the text and figures where clarifications are needed and the presentation could be improved. I recommend that the authors address all of these points before the revised paper is considered for publication in *Atmospheric Measurement Techniques*.

**Specific Comments**

**Title**

Lines 1–2. It's unclear what the measurement technique is from the title. Perhaps the words 'microwave (or, millimetre-wave) radiometer' could be included in the title?

**2.1 CORAM and Figure 1**

The technical description of the new instrument is difficult to follow and lacks important details about the novel electronic/optical components. Figure 1 is an unclear, poor quality diagram. It may be better to have two instrument diagrams, one showing the optical layout including the atmospheric and calibration load beam paths, and the other showing the electronic signal chain. For the atmospheric view, why was a 20° elevation chosen and what is the azimuthal angle? What is the field-of-view of the instrument and how far away from the instrument is the observed region of the middle atmosphere? What type of cryocooler is used? How much improvement in SNR is achieved by amplifying the 230 GHz signal by the first-stage LNA, rather than amplification occurring after down conversion to a lower frequency? Has this type of direct amplification been used before? For the FFTS, what alias is used, what is the frequency resolution, and why is this high spectral resolution needed? What baseline SNR is achieved in the recorded spectra? What are the integration times for the calibration and atmospheric signal measurements? Is the instrument located outdoors or inside a building and, if the latter, what external window material was used to transmit the atmospheric signal? What effects do local weather conditions have on the ground-based observations?

**Technical Corrections**

**Abstract**

Page 1, line 10. 'CO emissions'. It should be clarified that this refers to CO microwave (or, millimetre-wave) line emissions rather than, e.g. CO emissions from wildfires or industrial production.
Page 1, line 16. The exact start and end dates of the new dataset should be given.

**1. Introduction**

Page 1, lines 18–23. The authors should make it clear whether they are referring just to microwave (or, millimetre-wave) radiometers or also to other instruments operating at long-wave frequencies to measure thermal emissions from atmospheric molecules. If by 'electronic manipulation' the authors mean the use of heterodyne techniques then that should be clearly stated. Similarly, if 'reliance on the sun' is referring to solar absorption / occultation measurements then that should be made clear.
Page 1, line 26. 'An example of this…' An example of what?
Page 1, line 30. 'allows for more CO-poor air…' should probably be changed to 'allows more CO-poor air…'
Page 2, line 14. 'on smaller timescales…' Smaller than what?
Page 2, line 14. What type of waves are being referred to here?
Page 2, lines 16–17. 'relatively high time resolution'. Relative to what?
Page 2, lines 19–21. A time resolution of ≤1 hr may make the technique well-suited to observe periodic fluctuations in CO. However, the authors should consider how the limited resolution in vertical and horizontal directions would impact on observing structures on varying spatial scales associated with gravity waves, other dynamical processes, and the vortex edge.
Page 2, line 24. The authors should justify why the measurements provide a 'needed increase in Arctic coverage and an excellent opportunity…' How well placed are the Kiruna and Ny-Ålesund instruments for observations near the vortex edge and inside the winter-time Arctic polar vortex?
Page 2, line 25. A reference should be given to sudden stratospheric warmings.

**2. Instrument and measured data**

2.1 CORAM

Page 3, line 4. Define 'AWIPEV'.

2.2 Inversion method

Page 4, line 7. Define 'WACCM4'.
Page 4, line 7. Presumably data at the WACCM4 grid-point closest to the CORAM observations are used?
Page 4, line 8. '132-layer grid between approximately ground and 130 km altitude'. Why is this a 132-layer (altitude) grid rather than 131 layers (i.e., 0–1 km, 1–2 km, …, 130–131 km)?
Page 4, line 12. 'CO emissions are attenuated by water vapour in the atmosphere'. It should be clarified that attenuation is due to water vapour absorption of the CO signal. Presumably most of the water vapour is in the troposphere?
Page 4, line 14. '$O_3$ spectral line lies at 231.28 GHz…' The $O_3$ line position (i.e. line centre) is at 231.28 GHz.
Page 4, line 15. 'The spectroscopic line data used here is from…' should be changed to 'The spectroscopic line data used here are from…'
Page 4, line 22. 'ECMWF information is available four times per day'. Rephrase to remove any ambiguity, i.e. to make it clear that the ECMWF data are at six hour intervals.
Page 4, line 23. Change 'The temperature data is smoothed…' to 'The temperature data are smoothed…'
Page 4, lines 30–31. Suggest shorten 'Three primary sinusoids were found to be present, …' to 'Three primary sinusoids were found …'
Page 4, line 32. 'large compared with the width of the CO spectral line'. What is the width of the CO spectral line?
Page 5, line 2. Change 'estimated uncertainties of 1 and 0.5 K respectively' to 'estimated uncertainties of 1 K and 0.5 K respectively'.
Page 5, line 5. 'as a fraction of the a priori'. Which a priori?
Page 5, line 5. Change 'due the strong gradients in atmospheric CO' to 'due to the strong gradients in atmospheric CO'. Are the CO gradients in the horizontal or vertical direction, or both?

2.3 CO profile characteristics

Page 5, line 8. 'The instrument required maintenance after this date…' After which date?
Page 5, line 9. The unsmiley symbol on this line should probably be removed.

2.4 CO profile error estimates

Page 6, lines 5–6. 'An uncertainty of 1° is chosen for the pointing of the instrument to the sky, an overestimate of the motor uncertainty…' Could the actual pointing of the instrument be measured rather than relying on an output of the motor positioning mechanism? What is the motor referred to here?
Page 6, line 11. 'uncertainty in the line position is ignored because the frequency grid used in the inversion can be shifted …' Presumably adjusting the frequency grid deals with Doppler line-shifts as well as uncertainty in the line position? Perhaps the wording should be 'is shifted' rather than 'can be shifted'?

**3. Comparison with Aura MLS**

Page 6, line 24. 'the upper limit of the MLS CO retrieval altitude …' At what altitude is the upper limit?
Page 6, line 25. 'The data has a positive bias in the middle atmosphere, compared to the ACE-FTS satellite instrument, of 20% …' Define 'ACE-FTS'. Change 'The data has …' to 'The data have …' Do you mean that the MLS CO VMR data are 20% higher than the corresponding ACE-FTS data?
Page 6, line 26. 'Pumphrey et al., 2007'. The reference is missing.
Page 6, line 26. 'subsequent versions showing a slight decrease in the CO VMR.' State what are the subsequent MLS CO data versions. Do you mean a slight decrease in CO VMR values or a decrease in the CO VMR bias compared to ACE-FTS? Or perhaps both?

3.1 Colocated measurement comparison

Page 6, line 28. 'MLS measurements are subset to within ±2° latitude and ±10° longitude of CORAM'. Does this latitude/longitude range cover the location of the instrument on the ground and/or the CORAM observations in the middle atmosphere some distance away?
Page 6, lines 28–29. 'The CO VMRs are expected to vary more in latitude than in longitude.' Why is this expected?
Page 6, line 29. 'A longitude space of ±5° was also tested …' should probably be 'A longitude space of ±5° was tested …'
Page 6, line 30 – Page 7, line 1. 'Above 0.001 hPa, MLS CO profiles use a constant VMR value' should be 'Above 0.001 hPa, MLS CO profiles are constant in VMR value' or similar wording.
Page 7, line 10. 'mid-November to mid-January'. Please give the specific dates.
Page 7, line 18. 'The correlation between KIMRA and MLS was slightly higher, …' Slightly higher than the correlation between which other instruments?
Page 7, lines 21–22. 'after which the values become closer in VMR.' Please clarify - do you mean the MLS and CORAM profiles are in better agreement?
Page 7, lines 29 and 31. 'around December 22$^{nd}$, leading to a local minimum in the first week of January' and 'for about the first 25 hours'. I wonder if the authors could be more exact in the timings?

**4. CORAM data and usage**

Page 7, line 29. 'decrease in middle-atmospheric CO' should be 'decrease in middle-atmospheric CO VMR'.
Page 8, lines 2–3. 'Over this same time, between 60 and 70 km, there is an oscillation in the 4.1 and 6 ppmv contour lines, with peaks occurring every 1-2 hours.' Please could the authors provide some discussion of possible causes of the observed oscillation.
Page 8, lines 23–24. 'providing the averaging kernels do not significantly change over this time, which would change the measurement response.' Are the averaging kernels likely to change with time, and what might cause such changes?

**Conclusion**

Page 9, line 1. Suggest change 'CO profiles were retrieved …' to ''CO profiles were retrieved from observations …'
Page 9, lines 1–3. Suggest splitting this rather long sentence into two, e.g. with a full stop after '2017/2018' and starting the next sentence 'Error estimates…' It should be made clear that 'winter' refers to the northern hemisphere / Arctic. It would be worth restating in the conclusion the exact range of measurements dates.
Page 9, lines 6–7. 'abnormally high CO measured by CORAM above ~ 68 km in November' should be rewritten as 'abnormally high CO VMR measured by CORAM above ~ 68 km in November 2017'.

Page 9, lines 9–10.  'November 2017 to January 2018 are currently available.'  As suggested above, please give the exact dates for the dataset.  How can the available data be accessed?

**References**

The list of references appears to be sufficiently comprehensive and complete apart from the missing references for Pumphrey et al. (2007) and Kindlmann et al. (2002).  However, the list should be carefully checked and correctly formatted by the authors.

**Figures and Captions**

Figure 2.  Are the grid lines needed on the figures?  Figure 2(a) should be replotted with a minimum of ~300 K on the receiver noise temperature axis.

Figure 4.  'The measurement response (sum of the rows of the averaging kernels) divided by 4 is shown in solid blue.'  There are a number of lines in the plot coloured blue.  The colour scheme should be changed or the authors should make it clear whether the measurement response is shown by the thicker blue line.

Figure 5.  The axis label 'VMR [ppmv]' should be 'CO VMR [ppmv]'.

Figure 6.  The axis labels 'VMR' and '∆ VMR' should be 'CO VMR [ppmv]' and '∆ CO VMR [ppmv]' respectively.  For Figure 6(d) the correlation scale needs to be changed to make better use of the plot, e.g. the range from 0.7 to 1.0.

Figure 7.  Why were the selected altitudes chosen for plotting the time series?  The axis labels 'VMR [ppmv]' should be 'CO VMR [ppmv]'.  For Figure 7(a) the CO VMR scale should be adjusted to make better use of the plot, e.g. from 12 ppmv to 40 ppmv.

Figure 8.  Perhaps the main plot might be clearer with the data gaps shown in white rather than black?  Otherwise as presented the narrow black lines due to small data gaps look rather similar to the contour lines.  The colourbar labels 'VMR [ppmv]' should be 'CO VMR [ppmv]'.  Why were the particular CO VMR values chosen for the contours?  The reference to Kindlmann et al. (2002) is missing.  What is causing the gaps in the data record?

---

## Referee Comment (RC2) · Anonymous Referee #2 · 19 Apr 2019

This manuscript discusses middle atmospheric CO measurements carried out by a novel ground-based microwave spectrometer, CORAM, installed at the Arctic station of Ny-Ålesund (78.9° N, 11.9° E). The development of this instrument and its dataset are of interest to the scientific community, as CO is a useful tool for studying mesospheric dynamics in Polar regions and the satellite coverage of CO will become scarce in the near future. In fact, the creation of a network of ground-based instruments observing middle atmospheric constituents is desirable.

The paper is well written and well organized and I recommend this work be published. In my opionion, however, since this is the presentation paper for CORAM, there are

a few aspects of the instrumentation and the data presented that should be better discussed in the manuscript.

General comments

The paper lacks information on the receiver itself, possibly a photo, a sketch of the quasi-optical front end, and on the observing equations of this (total power?) instrument.

As a validation paper presenting a new receiver to the scientific community, I would expect there would be more data to show and that the validation would cover a longer time period. Especially since Polar mesospheric CO changes substantially from winter to summer, as do the observing capabilities of a 230 GHz ground-based instrument installed at sea level, so the data and their analysis results and uncertainties may change significantly from winter to summer. I understand that a technical failure occurred in January 2018 but now more than 14 months have passed. Are there new data to add to the analysis?

Specific comments

See the attached pdf file.

Please also note the supplement to this comment:
https://www.atmos-meas-tech-discuss.net/amt-2019-121/amt-2019-121-RC2-supplement.pdf

---

## Author Comment (AC1) · 20 Jun 2019

**Authors' response to referee comments**

We would like to take the opportunity to thank the referees for their time, and for their valuable feedback on the manuscript. We believe that their input has helped us to improve the manuscript where possible.

**Response to comments from Referee #1**

**General Comments**

The paper presents carbon monoxide (CO) measurements in the polar upper stratosphere and mesosphere made using a new ground-based millimetre-wave radiometer (CORAM). Atmospheric observations recorded during the 2017–18 winter from Ny-Ålesund are analysed using optimal estimation retrieval techniques to determine vertical profiles of CO volume mixing ratio. The precision of the measurements is estimated and CORAM profiles are compared with overlapping CO measurements by the Aura MLS satellite instrument.

Measurements of CO in the polar middle atmosphere are important as the gas is sufficiently long-lived to be used as a tracer for characterising dynamical and transport processes associated with the winter-time polar vortex, and atmospheric wave and tide activity. The structure and extent of the polar vortex above Antarctica, and its more dynamic counterpart in the Northern hemisphere affect global circulation patterns, stratospheric ozone abundances, and atmospheric heating rates. Targeted ground-based observations of CO, such as those presented here, complement the much larger geographical coverage of satellite remote sensing datasets. The ground-based instruments provide continuous observations with the potential to resolve features occurring on short timescales.

Overall the paper is reasonably well written and presented, with adequate description of the observations, data-sets, discussion of the results, and citing of prior work. The paper covers the development, inter-comparison, and validation of an atmospheric measurement system and this fits well with the subject areas of the journal. My main criticism is the lack of important details about the CORAM instrument itself. The main novelty in the work is the new design of this radiometer that incorporates optical/electronic components. The authors suggest that this design improves the performance/cost over previous heterodyne radiometers used for this type of measurement. As indicated in my specific comments below, the sections (2.1 and Figure 1) of the paper describing the new instrument need expanding with further technical details.

The manuscript would benefit from a more thorough description of existing CO measurement systems including those utilising the thermal IR bands of CO to make middle atmosphere observations. The conclusions section should include clear statements as to how well the anticipated improvements in performance were achieved. I've also identified a number of areas in the text and figures where clarifications are needed and the presentation could be improved. I recommend that the authors address all of these points before the revised paper is considered for publication in Atmospheric Measurement Techniques.

**Specific Comments**

**Title**
**Lines 1–2. It's unclear what the measurement technique is from the title. Perhaps the words 'microwave (or, millimetre-wave) radiometer' could be included in the title?**
The title has been changed to read:

*"Ground-based millimetre-wave measurements of middle-atmospheric carbon monoxide above Ny Ålesund (78.9°N, 11.9° E)."*

**2.1 CORAM and Figure 1**
**The technical description of the new instrument is difficult to follow and lacks important details about the novel electronic/optical components. Figure 1 is an unclear, poor quality diagram. It may be better to have two instrument diagrams, one showing the optical layout including the atmospheric and calibration load beam paths, and the other showing the electronic signal chain.**
Figure 1 has been remade for clarity and now includes a simplified version of the quasioptical system showing the relevant components. The caption of Figure 1 has been edited to reflect the changes. Combined with the expanded information in Section 2.1, the reader has a concise view of the instrument.

*"After the pointing mirror, the atmospheric signal is directed by a series of quasioptical components through a window in a cryocooler and fed into a corrugated horn antenna. The signal is amplified by a 230 GHz LNA. The unwanted sideband at ~ 213.5 GHz is supressed with a waveguide filter before the signal is mixed with the local oscillator (LO) signal (111 GHz) using a sub-harmonic mixer. Now at an intermediate frequency of 8.5 GHz, the signal exits the cooler and is amplified with another LNA before being further*

*downconverted to 0.5 GHz and analysed by a Fast Fourier Transform Spectrometer (FFTS). Figure 1 shows a schematic drawing of the receiver including the components in the cryocooler, as well as a simplified version of the quasioptical layout. The cryocooler makes use of a CTI Cryogenics 350 CP coldhead and a CTI Cryogenics 8200 compressor, as well as a helium cooling machine."*

*"Fig 1: Schematic of the CORAM receiver and simplified quasioptics. A rotatable mirror selects a signal from either the atmosphere, warm target, or cold target. The signal is directed by a parabolic mirror to a path length modulator that comprises a polarising wire grid, an absorber, and an oscillating rooftop mirror. The signal passes through a window in the cryocooler where it is directed to the receiver with an elliptical mirror. The signal enters the corrugated feed horn and encounters the RF LNA, a waveguide filter (BPF), and a sub-harmonic mixer (SHM). At the SHM the signal is downconverted to an intermediate frequency (IF) of 8.5 GHz. The IF signal exits the cryocooler and passes through a room temperature LNA. The RF (atmospheric) signal is mixed at the SHM with a local oscillator (LO), which is an 18.5 GHz signal from a phase-locked dielectric resonator oscillator (PDRO) that is passed through a x6 frequency multiplier, to provide 111 GHz. The IF out signal will be further downconverted to 0.5 GHz before being analysed by the Fast Fourier Transform Spectrometer (not shown here). Further details on quasioptical components can be found in Goldsmith (1998)."*

**For the atmospheric view, why was a 20° elevation chosen and what is the azimuthal angle?**
Motivation for the choice of viewing angle is now included in Section 2.1. The number is actually 21 degrees and the azimuth is 113 degrees:

*"The atmospheric signal enters the lab through a window that is transparent to millimetre-wave frequencies, and meets the pointing mirror of CORAM, angled at 21° elevation. This angle was chosen by performing a series of atmospheric radiative transfer simulations at different elevation angles, using a climatological polar winter atmosphere, and determining which angle provided the strongest CO spectral line. The choice of angle is a trade-off of maximum path length through the target gas in the atmosphere, and minimum attenuation of the target signal by atmospheric water vapour that is primarily in the troposphere. The azimuth angle of the atmospheric signal is 113°, defined by the lab in which CORAM is held."*

**What is the field-of-view of the instrument and how far away from the instrument is the observed region of the middle atmosphere?**

The HPBW is now included in Section 2.1.

*"The quasioptical setup has an antenna pattern with a half-power-beam-width of ~ 5°."*

With an elevation angle of 21 degrees, the distance from the instrument can be calculated at any altitude. The distance at 60 km is now included in Section 3.1.

*"MLS measurements are subset to within $\pm 2°$ latitude and $\pm 10°$ longitude of CORAM, calculated at 60 km altitude along the line of sight of CORAM (~ 156 km horizontally from the lab)."*

**What type of cryocooler is used?**

Section 2.1 now contains the following information:

*"The cryocooler makes use of a CTI Cryogenics 350 CP coldhead and a CTI Cryogenics 8200 compressor, as well as a helium cooling machine."*

**How much improvement in SNR is achieved by amplifying the 230 GHz signal by the first-stage LNA, rather than amplification occurring after down conversion to a lower frequency?**

Information is now included that describes an estimate of the difference in the receiver temperature when switching the position of the LNA relative to the mixer, as well as an outline of the radiometer equation, which describes how the SNR is related to the system temperature. The system temperature includes more contributions than just the receiver temperature of CORAM, which is mainly from the LNA/filter/mixer. The SNR is not a fixed value and will change depending on the atmospheric conditions. This information is now clarified in Section 2.1.

*"An estimate of the improvement in the receiver temperature (Janssen, 1993) can be made using a noise temperature cascade analysis. A variation of Friis' equation (Vowinkel, 1988) for two components in succession is $T = T_1 + T_2/G_1$, where $T_1$, and $T_2$ are the respective noise temperatures of the first and second components, $G_1$ is the linear gain of the first component, and T is the total noise temperature. The noise temperature of the LNA plus waveguide filter was measured to be 1350 K at room temperature, and the linear gain was measured at 158 (corresponding to 22 dB) (Fig. 2b). The noise temperature of the sub-harmonic mixer is ~1500 K at room temperature and has a linear gain of ~0.16*

(corresponding to -8 dB). Applying Friis' equation with the LNA preceding the mixer gives a noise temperature of ~1360 K. The same calculation with the mixer as the first component gives a noise temperature of ~9800 K. The dominant contribution to the receiver temperature of CORAM is from the LNA/filter/mixer. Cooling the components can considerably reduce their noise temperature. Figure 2b shows the noise temperature and gain of the LNA + filter, measured at room temperature. Figure 2c shows the receiver temperature for CORAM measured at the exit of the cryocooler, with the cryocooler components at a typical temperature of 39 K. At 8.5 GHz, the receiver temperature is below 350 K.

The system temperature, $T_{sys}$, includes contributions the second downconversion, the atmospheric background and signal, and quasioptical spillover (Parrish et al., 1988, Janssen, 1993, Stanimirović et al., 2002). The system temperature is related to the measurement time through the so-called ideal radiometer equation: $\sigma_T = T_{sys} / (Bt)^{1/2}$, where $\sigma_T$ is the statistical noise on a measured spectrum, B is the frequency bandwidth of the measurement, and t is the integration time for the measurement. This relationship determines the measurement time required to provide a given SNR. The single sideband $T_{sys}$ for CORAM is ~600 K. The atmospheric measurements are all made with the same elevation angle and so the individually recorded spectra can be averaged together to reduce the SNR. The measurements used here have been spectrally averaged over approximately 1 hour, including time used to calibrate the signal. Finer time resolutions that still yield usefully high SNRs are possible. Since $T_{sys}$, as defined here, contains a component from the atmospheric background, the SNR of a given measurement will vary with the atmospheric conditions at the time, with a more opaque troposphere giving rise to a smaller SNR. An ad-hoc indication of "bad" weather conditions was found to be a measurement with a baseline temperature > 230 K, and these measurements were discarded."

**Has this type of direct amplification been used before?**
This configuration has been used for other radiometers that operate at lower frequencies. The following information has been included in Section 2.1. The new citations have been included in the reference list.
"This configuration has been used before for similar instruments e.g. MIAWARA-C (Straub et al. 2010) and GROMOS-C (Fernandez et al., 2015), which measure ozone at 110 GHz, and water vapour at 22 GHz, respectively."

**For the FFTS, what alias is used, what is the frequency resolution, and why is this high spectral resolution needed?**

With a bandwidth of 1 GHz and 16384 channels the FFTS provides a frequency resolution of about 61 kHz. A higher frequency resolution, depending on the Doppler width, is required for resolving the mesospheric part of the spectrum. The following text is now included in Section 2.1

*"The FFTS is an Acqiris AC240 and has a bandwidth of 1 GHz with 16384 channels, providing ~61 kHz resolution. A high spectral resolution, depending on the Doppler width of a spectral line (~300 kHz in this case), is required for resolving the mesospheric contribution to the spectrum. CORAM performs the Fourier Transform in real time and the fully resolved spectrum is stored."*

It is not clear what is meant by "what alias is used?".

**What baseline SNR is achieved in the recorded spectra?**
The SNR of a particular measurement is governed by the system temperature, which includes an atmospheric component. The weather is quite variable at Ny Alesund, and this changes the SNR from measurement to measurement. The equation that describes the relationship between the SNR and the system temperature is now given in Section 2.1.

*"The system temperature, $T_{sys}$, includes contributions from the second downconversion, the atmospheric background and signal, and quasioptical spillover (Parrish et al., 1988, Janssen, 1993, Stanimirović et al., 2002). The system temperature is related to the measurement time through the so-called radiometer equation: $\sigma_T = T_{sys} / (Bt)^{1/2}$, where $\sigma_T$ is the statistical noise on a measured spectrum, B is the frequency bandwidth of the measurement, and t is the integration time for the measurement. This relationship determines the measurement time required to provide a given SNR. The single sideband $T_{sys}$ for CORAM is ~600 K. ……… . Since $T_{sys}$, as defined here, contains a component from the atmospheric background, the SNR of a given measurement will vary with the atmospheric conditions at the time, with a more opaque troposphere giving rise to a smaller SNR. An ad-hoc indication of "bad" weather conditions was found to be a measurement with a baseline temperature > 230 K, and these measurements were discarded."*

**What are the integration times for the calibration and atmospheric signal measurements?**
CORAM currently measures each target with equal integration times. An error propagation of the total power formula shows, that the noise on the calculated spectrum depends on the time the calibration blackbodies are measured and on the level the target signal (here: the atmosphere) has. As a compromise, equal measuring times for all sources have been

chosen. The data used here has been spectrally averaged over a 1 hour period including time for calibration.

Section 2.1
*"The measured atmospheric signal is calibrated using two blackbody targets at known temperatures (measured with mounted sensors): a cold target in the cryocooler at ~ 70 K and a warm target at ~ 293 K. The integration times for each blackbody is the same as that for the atmospheric signal."*

*"The measurements used here have been spectrally averaged over approximately 1 hour, including time used to calibrate the signal."*

**Is the instrument located outdoors or inside a building and, if the latter, what external window material was used to transmit the atmospheric signal?**
The instrument is located inside a lab and the atmospheric signal enters through a foam window that is transparent to millimetre-wave frequencies. This information is now included in Section 2.1. Investigation of possible attenuation of the atmospheric signal by the window has been included in Section 5: Conclusions and future work.

Section 2.1
*"The atmospheric signal enters the lab through a window that is transparent to millimetre-wave frequencies, and meets the pointing mirror of CORAM, angled at 21° elevation."*

Section 5
*"Future work with CORAM will include: Integration of a new local oscillator due to a failure of the original, and investigation of possible attenuation of the atmospheric signal by the laboratory foam window."*

**What effects do local weather conditions have on the ground-based observations?**
Local weather conditions can cause some measurements to be unusable. An ad-hoc indication of "bad" weather conditions was found to be a measurement with a baseline temperature > 230 K, and these measurements were discarded. This information is now included at the end of Section 2.1.

**Technical Corrections**

**Abstract**
**Page 1, line 10. 'CO emissions'. It should be clarified that this refers to CO microwave (or, millimetre-wave) line emissions rather than, e.g. CO emissions from wildfires or industrial production.**
The line has been edited to read *"… spectral emissions …"*

**Page 1, line 16. The exact start and end dates of the new dataset should be given.**
This has been added.

**1. Introduction**
**Page 1, lines 18–23. The authors should make it clear whether they are referring just to microwave (or, millimetre-wave) radiometers or also to other instruments operating at long-wave frequencies to measure thermal emissions from atmospheric molecules. If by 'electronic manipulation' the authors mean the use of heterodyne techniques then that should be clearly stated. Similarly, if 'reliance on the sun' is referring to solar absorption / occultation measurements then that should be made clear.**
The wording has been changed to include "milimetre-wave" and refer to solar absorption measurements. The latter part has been expanded to clarify the benefits of this type of instrumentation, including a reference to coherent detection with heterodyne receivers. A citation for Jannsen (1993), has been added for further reading.

*"Millimetre-wave (also referred to as microwave) radiometers are powerful tools for measuring the composition of the atmosphere. This is particularly true for areas where there are prolonged night-time periods, such as the poles. The instruments can measure emissions from molecules in the atmosphere, in contrast to solar absorption measurements that rely on the sun. Coherent detection of the atmospheric signal, achieved through heterodyne receivers, and electronic manipulation of that signal, make it possible to detect and resolve spectral lines with very low intensities, especially when the electronics are cooled to low temperatures, thus producing lower thermal noise (Janssen, 1993)."*

**Page 1, line 26. 'An example of this…' An example of what?**
The sentence has been changed to the following:

*"During polar night, CO concentrations increase in the middle atmosphere due to the vertical branch of the residual mean circulation bringing CO-rich air from higher altitudes (Smith et al., 2011; Garcia et al., 2014)."*

**Page 1, line 30. 'allows for more CO-poor air…' should probably be changed to 'allows more CO-poor air…'**
This has been changed.

**Page 2, line 14. 'on smaller timescales…' Smaller than what?**
The sentence directly before this one refers to variation in VMR on a timescale of days: *"+450 m/day'.* The next sentence (the one in question) refers to variations on timescales smaller than this, explicitly stated as *"minutes to hours"*.

**Page 2, line 14. What type of waves are being referred to here?**
No distinction is made here. The five references offer analyses on the types of waves that cause disturbances on these time scales.

**Page 2, lines 16–17. 'relatively high time resolution'. Relative to what?**
The sentence has been edited to read:

*"Data from ground-based radiometers with high time resolution (order of an hour or less) have be used to investigate small periodic fluctuations in ozone ($O_3$) and water vapour (Hocke et al., 2006; Moreira et al., 2018, Schranz et al., 2018)."*

**Page 2, lines 19–21. A time resolution of ≤1 hr may make the technique well-suited to observe periodic fluctuations in CO. However, the authors should consider how the limited resolution in vertical and horizontal directions would impact on observing structures on varying spatial scales associated with gravity waves, other dynamical processes, and the vortex edge.**
This future work will most likely be performed using techniques, or variations thereof, that have been used in the cited literature. The measurements in these citations come from both ground-based and satellite-borne instruments with varying degrees of spatial resolution. As such, the paragraph in the manuscript has been edited to include the following:

*"As with the ground-based and satellite-borne instruments in the works cited above, the analyses must be performed within the context of the limited spatial resolution of the measurements."*

**Page 2, line 24. The authors should justify why the measurements provide a 'needed increase in Arctic coverage and an excellent opportunity…' How well placed are the Kiruna and Ny-Ålesund instruments for observations near the vortex edge and inside the winter-time Arctic polar vortex?**

Information has been added on the sparsity of recent polar datasets to justify the need for more.

*"CO profiles from satellite measurements have been used regularly to study processes in the polar winter atmosphere (e.g. Damiani et al., 2014; Lee et al., 2011; Manney et al., 2009; McLandress et al., 2013), but recent ground-based CO datasets in the polar (and nearby) regions have been sparse: The Onsala Space Observatory instrument (57°N, 12°E) (Forkmann et al., 2012), which produced data for 2002 – 2008, and from 2014; The ground-based millimetre-wave spectrometer (GBMS) at Thule Air Base (76.5°N, 68.7° W), used to investigate the Arctic winter of 2001/2002 (Muscari et al., 2007) and the sudden stratospheric warming (SSW) in 2009 (Di Biagio et al., 2010); The British Antarctic Survey (BAS) radiometer data at Troll Station (72°S, 2.5°E) covers February 2008 to January 2010 (Straub et al., 2013). These instruments also measure the rotational transitions of CO and can operate during polar night."*

The vortex is not a stable structure in space, but having two instruments operating at the same time that are 12 degrees apart within the polar region will provide opportunities to measure inside/outside/in the edge of the polar vortex.

**Page 2, line 25. A reference should be given to sudden stratospheric warmings.**

Sudden stratospheric warmings are now mentioned earlier, in Section 1, in relation to the citation of Di Biagio et al. (2010).

**2. Instrument and measured data**
**2.1 CORAM**
**Page 3, line 4. Define 'AWIPEV'.**

AWIPEV is the name of the research base.

**2.2 Inversion method**
**Page 4, line 7. Define 'WACCM4'.**

This is now defined as the Whole Atmosphere Community Climate Model.

**Page 4, line 7. Presumably data at the WACCM4 grid-point closest to the CORAM observations are used?**

A line is added directly after to clarify:

*"Model output for the grid point encompassing Ny-Ålesund is used."*

**Page 4, line 8. '132-layer grid between approximately ground and 130 km altitude'. Why is this a 132-layer (altitude) grid rather than 131 layers (i.e., 0–1 km, 1–2 km, …, 130–131 km)?**

The grid is in pressure space. It does not follow altitude in such a fashion. The sentence has been edited to clarify that the grid is in pressure space:

*"The output is on a 132-layer pressure grid between approximately ground and 130 km altitude."*

**Page 4, line 12. 'CO emissions are attenuated by water vapour in the atmosphere'. It should be clarified that attenuation is due to water vapour absorption of the CO signal. Presumably most of the water vapour is in the troposphere?**

The sentence has been edited to clarify:

*"CO emissions are attenuated by absorption due to water vapour in the atmosphere (mostly in the troposphere) and this is accounted for by including the water vapour continuum by Rosenkranz (1998) in the forward model and inversion."*

**Page 4, line 14. '$O_3$ spectral line lies at 231.28 GHz…' The $O_3$ line position (i.e. line centre) is at 231.28 GHz.**

The sentence has been edited to clarify that the centre of the spectral line is at 231.28 GHz.

*"$O_3$ is also simultaneously retrieved with CO, as an $O_3$ spectral line is centred at 231.28 GHz."*

**Page 4, line 15. 'The spectroscopic line data used here is from…' should be changed to 'The spectroscopic line data used here are from…'**

This has been fixed.

**Page 4, line 22. 'ECMWF information is available four times per day'. Rephrase to remove any ambiguity, i.e. to make it clear that the ECMWF data are at six hour intervals.**

This is now clarified.

*"ECMWF information is available daily at 6-hour intervals, beginning at midnight, and covers up to 0.01 hPa altitude"*

**Page 4, line 23. Change 'The temperature data is smoothed…' to 'The temperature data are smoothed…'**
This has been fixed.

**Page 4, lines 30–31. Suggest shorten 'Three primary sinusoids were found to be present, …' to 'Three primary sinusoids were found …'**
Agreed. This has been changed.

**Page 4, line 32. 'large compared with the width of the CO spectral line'. What is the width of the CO spectral line?**
This information has been added to the line.

*"The periods of the sinewaves are large compared with the width of the CO spectral line, which has a typical full-width at half-maximum (FWHM) of ~0.7 MHz, and so are uniquely distinguishable from it."*

**Page 5, line 2. Change 'estimated uncertainties of 1 and 0.5 K respectively' to 'estimated uncertainties of 1 K and 0.5 K respectively'.**
This has been changed.

**Page 5, line 5. 'as a fraction of the a priori'. Which a priori?**
This has been edited to clarify that the CO VMR is retrieved as a fraction of the CO a priori.

*"CO VMRs are retrieved as a fraction of their a priori for numerical stability due the strong vertical gradients in atmospheric CO."*

**Page 5, line 5. Change 'due the strong gradients in atmospheric CO' to 'due to the strong gradients in atmospheric CO'. Are the CO gradients in the horizontal or vertical direction, or both?**
Vertical. This is now clarified in the line.

*"CO VMRs are retrieved as a fraction of their a priori for numerical stability due to the strong vertical gradients in atmospheric CO."*

**2.3 CO profile characteristics**
**Page 5, line 8. 'The instrument required maintenance after this date…' After which date?**
The sentence now indicates that it is the latter date in January.

**Page 5, line 9. The unsmiley symbol on this line should probably be removed.**
The emoji adds a culturally independent levity to the writing, without detracting from the scientific detail.

**2.4 CO profile error estimates**
**Page 6, lines 5–6. 'An uncertainty of 1° is chosen for the pointing of the instrument to the sky, an overestimate of the motor uncertainty…' Could the actual pointing of the instrument be measured rather than relying on an output of the motor positioning mechanism? What is the motor referred to here?**
The elevation angle of the instrument is measured at 21 degrees. This is indicated in Section 2.1. The sentence has been edited to clarify that the overestimation is to account for changes that might occur in the orientation of instrument table. The model of the motor is now included in the edited manuscript.
*"An uncertainty of $1^\circ$ is chosen for the pointing of the instrument to the sky, an overestimate of the motor (Faulhaber 3564K024B CS) uncertainty by an order of magnitude, to account for changes that may occur the orientation of the instrument table."*

Section 2.1 now includes information on the measurement of the elevation angle.
*"The alignment of the quasioptical components was checked using a laser positioned at the entrance to the cryocooler. The elevation angle of the instrument was measured using a self-levelling laser (Bosch GLL 3-80), which provides a horizontal line with an accuracy of 0.2 mm/m (0.2 mrad). Two horizontal lines, one directly from the laser and one passing through the quasioptical setup, were aligned on a screen approximately 5 m from the instrument. A sun scanning method has been used with other ground-based instruments to identify a pointing offset, e.g., for MIAWARA-C (Straub et al. 2010) and GROMOS-C (Fernandez et al., 2015), for which the offsets in the elevation angle were found to be $0.01^\circ$ and $0.07^\circ$, respectively."*

**Page 6, line 11. 'uncertainty in the line position is ignored because the frequency grid used in the inversion can be shifted …' Presumably adjusting the frequency grid deals with Doppler line-shifts as well as uncertainty in the line position? Perhaps the wording should be 'is shifted' rather than 'can be shifted'?**

Adjusting the frequency grid does not deal with doppler line shifts that are caused by winds at different altitudes, as they differ in direction and magnitude. CORAM does not have the spectral resolution to observe these changes. "can be shifted" has been changed to "is shifted".

**3. Comparison with Aura MLS**

**Page 6, line 24. 'the upper limit of the MLS CO retrieval altitude …' At what altitude is the upper limit?**

The preceding sentence states the pressure range of the data:

*"The atmospheric pressure range of the data is 215 - 0.0046 hPa."*

**Page 6, line 25. 'The data has a positive bias in the middle atmosphere, compared to the ACE-FTS satellite instrument, of 20% …' Define 'ACE-FTS'. Change 'The data has …' to 'The data have …' Do you mean that the MLS CO VMR data are 20% higher than the corresponding ACE-FTS data?**

The sentence has been edited to clarify these points:

*"The data have a positive bias of 20% in the middle atmosphere (larger VMRs), compared to the Atmospheric Chemistry Experiment – Fourier Transform Spectrometer (ACE-FTS) satellite instrument (Livesey et al., 2015)."*

**Page 6, line 26. 'Pumphrey et al., 2007'. The reference is missing.**

The reference has been added.

**Page 6, line 26. 'subsequent versions showing a slight decrease in the CO VMR.' State what are the subsequent MLS CO data versions. Do you mean a slight decrease in CO VMR values or a decrease in the CO VMR bias compared to ACE-FTS? Or perhaps both?**

The sentence mentions MLS data version 2.2, so subsequent data versions means ones that are after version 2.2. Listing versions of MLS data is superfluous information for the

manuscript. An interested reader may find information on the MLS website for datasets that are not used in this work. The MLS data quality document also refers to "later versions". Extra information has been included in the edited manuscript here to clarify that the MLS CO data since v2.2 has decreased in magnitude in the middle atmosphere, bringing the values closer to ACE-FTS.

*"The data have a positive bias of 20% in the middle atmosphere (larger VMRs), compared to the Atmospheric Chemistry Experiment – Fourier Transform Spectrometer (ACE-FTS) satellite instrument (Livesey et al., 2015). This bias is estimated from a study of Version 2.2 of MLS CO data (Pumphrey et al., 2007), which showed a positive bias of 30 %. Subsequent versions of MLS CO, including the version used here, show a slight decrease in the CO VMR, bringing the values closer to those of ACE-FTS."*

**3.1 Colocated measurement comparison**
**Page 6, line 28. 'MLS measurements are subset to within ±2° latitude and ±10° longitude of CORAM'. Does this latitude/longitude range cover the location of the instrument on the ground and/or the CORAM observations in the middle atmosphere some distance away?**
The sentence has been edited to read:

*"MLS measurements are subset to within $\pm 2°$ latitude and $\pm 10°$ longitude of CORAM, calculated at 60 km altitude along the line of sight of CORAM (~ 156 km horizontally from the lab)."*

**Page 6, lines 28–29. 'The CO VMRs are expected to vary more in latitude than in longitude.' Why is this expected?**
The atmospheric composition generally varies more in the meriodional direction compared to the zonal direction. This information is added to the line.

*"The CO VMRs are expected to vary more in latitude than in longitude because the atmospheric composition generally varies more in the meridional direction compared to the zonal."*

**Page 6, line 29. 'A longitude space of ±5° was also tested …' should probably be 'A longitude space of ±5° was tested …'**
This has been changed.

**Page 6, line 30 – Page 7, line 1. 'Above 0.001 hPa, MLS CO profiles use a constant VMR value' should be 'Above 0.001 hPa, MLS CO profiles are constant in VMR value' or similar wording.**
This has been changed,

**Page 7, line 10. 'mid-November to mid-January'. Please give the specific dates.**
This has been changed to *"November 19th to January 18th."*

**Page 7, line 18. 'The correlation between KIMRA and MLS was slightly higher, …' Slightly higher than the correlation between which other instruments?**
It is now clarified:

*"The correlation between KIMRA and MLS was slightly higher than that for CORAM and MLS, remaining greater than 0.90 up to 82 km altitude."*

**Page 7, lines 21–22. 'after which the values become closer in VMR.' Please clarify - do you mean the MLS and CORAM profiles are in better agreement?**
This has been clarified:

*"The largest differences in CO are found at higher altitudes (≥68 km) in November and the first days of December, after which the values become closer in VMR, indicating better agreement between the instruments."*

**Page 7, lines 29 and 31. 'around December 22nd, leading to a local minimum in the first week of January' and 'for about the first 25 hours'. I wonder if the authors could be more exact in the timings?**
This sentence has been removed and more detail is put into the description of the plot in Figure 8 that is a subset of the data. Descriptions for this plot are clearer for the reader.

**4. CORAM data and usage**
**Page 7, line 29. 'decrease in middle-atmospheric CO' should be 'decrease in middle-atmospheric CO VMR'.**
This sentence has been removed (see response to previous comment) and more detail is put into the description of the plot in Figure 8 that is a subset of the data. Descriptions for this plot are clearer for the reader.

**Page 8, lines 2–3. 'Over this same time, between 60 and 70 km, there is an oscillation in the 4.1 and 6 ppmv contour lines, with peaks occurring every 1-2 hours.' Please could the authors provide some discussion of possible causes of the observed oscillation.**
The beginning of the next paragraph in same section includes possible causes of the observed oscillations and provides citations to works that have analyzed similar features:

*"These are broad descriptions of the data because one cannot fully characterise the variations in CO without the use of other data sources and model output. Variations on the timescales of an hour to weeks are visible in the data and require detailed study to elucidate the underlying dynamical processes, such as polar vortex shift, Rossby wave activity, SSW events, gravity wave perturbations (time scales of minutes to hours). Peridocities in trace gas data have previously been analysed using spectral decomposition techniques on ground-based measurements of water vapour and ozone (e.g., Struder et al., 2012, Hocke et al., 2013, Schranz et al., 2019) to identify waves with periods of days to weeks."*

**Page 8, lines 23–24. 'providing the averaging kernels do not significantly change over this time, which would change the measurement response.' Are the averaging kernels likely to change with time, and what might cause such changes?**
The next sentence has been expanded to clarify.

*"The measurement response for CORAM should not show significantly variation inside the retrievable altitude range but care should be taken at altitudes near the edges of the retrieval range of the profiles, where the measurement response has a strong gradient and can change quickly when there are rapid changes in CO concentrations at those altitudes."*

**Conclusion**
**Page 9, line 1. Suggest change 'CO profiles were retrieved …' to ''CO profiles were retrieved from observations …'**
*This has been edited to say "…retrieved from measurements…"*

**Page 9, lines 1–3. Suggest splitting this rather long sentence into two, e.g. with a full stop after '2017/2018' and starting the next sentence 'Error estimates…' It should be made clear that 'winter' refers to the northern hemisphere / Arctic. It would be worth restating in the conclusion the exact range of measurements dates.**
The sentence has been broken into two and the dates added.

*"CO profiles were retrieved from measurements in the Arctic winter of 2017/2018 (November 18th to January 19th). Error estimates show that the uncertainty in the temperature input for the inversions and the statistical noise on the spectrum are the largest contributions to the error budget, giving a maximum in the error profile of ~ 12 % of the a priori profile."*

**Page 9, lines 6–7. 'abnormally high CO measured by CORAM above ~ 68 km in November' should be rewritten as 'abnormally high CO VMR measured by CORAM above ~ 68 km in November 2017'.**
This has been added.

**Page 9, lines 9–10. 'November 2017 to January 2018 are currently available.' As suggested above, please give the exact dates for the dataset. How can the available data be accessed?**
The dates have been added.
There is now a section on data availability.

**References**
**The list of references appears to be sufficiently comprehensive and complete apart from the missing references for Pumphrey et al. (2007) and Kindlmann et al. (2002). However, the list should be carefully checked and correctly formatted by the authors.**

**Figures and Captions**
**Figure 2. Are the grid lines needed on the figures? Figure 2(a) should be replotted with a minimum of ~300 K on the receiver noise temperature axis.**
The grid lines are thinner than the plotted data values and are a different colour so as not to be confused with the data.
This plot was provided by engineers during the testing phase and we do not have the original data to replot.

**Figure 4. 'The measurement response (sum of the rows of the averaging kernels) divided by 4 is shown in solid blue.' There are a number of lines in the plot coloured blue. The colour scheme should be changed or the authors should make it clear whether the measurement response is shown by the thicker blue line.**
It is now clarified that the measurement response is the thick solid blue line.

**Figure 5. The axis label 'VMR [ppmv]' should be 'CO VMR [ppmv]'.**
This has been changed.

**Figure 6. The axis labels 'VMR' and 'Δ VMR' should be 'CO VMR [ppmv]' and 'Δ CO VMR [ppmv]' respectively. For Figure 6(d) the correlation scale needs to be changed to make better use of the plot, e.g. the range from 0.7 to 1.0.**
These changes have been made.

**Figure 7. Why were the selected altitudes chosen for plotting the time series? The axis labels 'VMR [ppmv]' should be 'CO VMR [ppmv]'. For Figure 7(a) the CO VMR scale should be adjusted to make better use of the plot, e.g. from 12 ppmv to 40 ppmv.**
The altitudes span the retrieval altitude range in equal spacing. The changes have been made.

**Figure 8. Perhaps the main plot might be clearer with the data gaps shown in white rather than black? Otherwise as presented the narrow black lines due to small data gaps look rather similar to the contour lines. The colourbar labels 'VMR [ppmv]' should be 'CO VMR [ppmv]'. Why were the particular CO VMR values chosen for the contours? The reference to Kindlmann et al. (2002) is missing. What is causing the gaps in the data record?**
These changes have been implemented. The reference for Kindlmann et al. (2002) has been added.

---

## Author Comment (AC2)

**Authors' response to referee comments**
We would like to take the opportunity to thank the referees for their time, and for their valuable feedback on the manuscript. We believe that their input has helped us to improve the manuscript where possible.

**Response to comments from Referee #2**

**This manuscript discusses middle atmospheric CO measurements carried out by a novel ground-based microwave spectrometer, CORAM, installed at the Arctic station of Ny-Ålesund (78.9°N, 11.9°E). The development of this instrument and its dataset are of interest to the scientific community, as CO is a useful tool for studying mesospheric dynamics in Polar regions and the satellite coverage of CO will become scarce in the near future.  In fact, the creation of a network of ground-based instruments observing middle atmospheric constituents is desirable. The paper is well written and well organized and I recommend this work be published. In my opionion, however, since this is the presentation paper for CORAM, there are  few  aspects  of  the  instrumentation  and  the data  presented  that  should  be  better discussed in the manuscript. General comments The paper lacks information on the receiver itself,  possibly a photo,  a sketch of the quasi-optical front end, and on the observing equations of this (total power?)  instrument. As a validation paper presenting a new receiver to the scientific community, I would expect there would be more data to show and that the validation would cover a longer time period. Especially since Polar mesospheric CO changes substantially from winter to summer, as do the observing capabilities of a 230 GHz ground-based instrument installed at  sea  level,  so  the  data  and  their  analysis  results  and  uncertainties  may change significantly from winter to summer.  I understand that a technical failure occurred in January 2018 but now more than 14 months have passed.  Are there new data to add to the analysis?**
The local oscillator broke down in January. The Element became unstable and changed its frequency randomly and with a low frequency. At the measurement site we lack equipment to diagnose such a failure and asked the manufacturer to help in the diagnosis.
The production of a new element took another 12 months, hence we will be able to start measurements again in September 2019.

**Specific comments**

**Page: 1**
**Sequence number: 1**
**Author:**
**Date: 12/04/2019 11:37:32**
**It's not clear what is intended with "precision" here. Would it be better to indicate the estimated total uncertainty instead?**
The wording has been changed to uncertainty because the value from the estimated uncertainty in the profile is used here.

**Page: 2**
**Sequence number: 1**
**Author:**
**Date: 12/04/2019 11:47:08**
**have been**
This has been fixed.

**Sequence number: 2**
**Author:**
**Date: 12/04/2019 11:51:35**
**The poor vertical resolution of the datasets could be a problem for studying gravity wave-induced fluctuations. Maybe a comment**
**on this aspect is needed.**
The introduction now refers to the limited spatial resolution of the cited ground-based and satellite-borne instruments that have been used to study periodic fluctuations in trace gas profiles.

"The positive gradient of polar CO VMRs with altitude throughout the middle atmosphere, coupled with the time resolution of the presented measurement system at Ny-Ålesund ($\leq 1$ hr), means that the dataset discussed here is well-suited to observing these periodic fluctuations, which are likely to be caused by vertical advection of air parcels by gravity waves (Zhu and Holton, 1997; Ekermann et al., 1998; Hocke et al., 2006). As with the ground-based and satellite-borne instruments in the works cited above, the analyses must be performed within the context of the limited spatial resolution of the measurements."

**Page: 3**
**Sequence number: 1**

**Author:**
**Date: 12/04/2019 12:55:04**
**I would provide a photo of the instrument to have an idea of its front-end and how it is installed.**
The photographs of the instrument do not offer clarity on the 3-dimensional optical bench. It is more likely to confuse the reader. A schematic of the front end in Figure 1 now includes the main quasioptical components and beam paths for the signal, hot target, and cold target.

**Sequence number: 2**
**Author:**
**Date: 19/04/2019 16:28:28**
**What materials were used for the window of the lab and the window of the cryocooler? Is it a total power instrument?**
This information is now included in Section 2.1.

*"CORAM is total-power radiometer housed at Ny-Ålesund, Svalbard (78.9°N, 11.9°E), and is part of the joint French-German Arctic Research Base, AWIPEV."*

*"The atmospheric signal enters the lab through a foam window that is transparent to millimetre-wave frequencies, and meets the pointing mirror of CORAM, …"*

*"After the pointing mirror, the atmospheric signal is directed by a series of quasioptical components through a mylar window in a cryocooler and fed into a corrugated horn antenna."*

**Sequence number: 3**
**Author:**
**Date: 19/04/2019 16:27:50**
**Authors should draw a sketch of this serie of mirrors, show how the signal is directed to the horn, and how they account for these
multiple reflections in their estimate of the elevation angle of their signal beam.**
Figure 1 has been edited and now contains a simplified version of the quasioptics that demonstrates how the signal enters the horn from the pointing mirror.

The alignment is checked using a laser positioned at the entrance to the cryocooler. Section 2.1 now contains this information.

*"Figure 1 shows a schematic drawing of the receiver including the components in the cryocooler, as well as a simplified version of the quasioptical layout The alignment of the quasioptical components was checked using a laser positioned at the entrance to the cryocooler. The elevation angle of the instrument was measured using a self-levelling laser (Bosch GLL 3-80), which provides a horizontal line with an accuracy of 0.2 mm/m (0.2 mrad). Two horizontal lines, one directly from the laser and one passing through the quasioptical setup, were aligned on a screen approximately 5 m from the instrument. A sun scanning method has been used with other ground-based instruments to identify a pointing offset, e.g., for MIAWARA-C (Straub et al. 2010) and GROMOS-C (Fernandez et al., 2015), for which the offsets in the elevation angle were found to be 0.01° and 0.07°, respectively."*

**Sequence number: 4**
**Author:**
**Date: 12/04/2019 14:38:32**
**Does it enter the FFTS at 1.5 GHz?**
**Later on you write that the FFTS is the AC240 with a 1 GHz bandwidth, therefore the signal enters the FFTS at 500 MHz I guess. Is**
**this correct?**
Yes, thank you. This has been fixed.

**Sequence number: 5**
**Author:**
**Date: 12/04/2019 14:19:59**
**Why not writing the equation Tnoise = T1+T2/G1+T3/G4+…**
This equation is now included in Section 2.1 to provide an estimate of the difference in noise temperature that comes with having an amplifier before the mixer.

*"An estimate of the improvement in the receiver temperature (Janssen, 1993) can be made using a noise temperature cascade analysis. A variation of Friis' equation (Vowinkel, 1988) for two components in succession is $T = T_1 + T_2/G_1$, where $T_1$, and $T_2$ are the respective noise temperatures of the first and second components, $G_1$ is the linear gain of the first component, and T is the total noise temperature. The noise temperature of the LNA plus waveguide filter was measured to be 1350 K at room temperature, and the linear gain was*

*measured at 158 (corresponding to 22 dB) (Fig. 2b). The noise temperature of the sub-harmonic mixer is ~ 1500 K at room temperature and has a linear gain of ~ 0.16 (corresponding to -8 dB). Applying Friis' equation with the LNA preceding the mixer gives a noise temperature of ~ 1360 K. The same calculation with the mixer as the first component gives a noise temperature of ~ 9800 K. The dominant contribution to the noise temperature of CORAM is from the LNA/filter/mixer. Cooling the components can considerably reduce their noise temperature. Figure 2b shows the noise temperature and gain of the LNA + filter, measured at room temperature. Figure 2c shows the receiver temperature for CORAM measured at the exit of the cryocooler, with the cryocooler components at a typical temperature of 39 K. At 8.5 GHz, the receiver temperature is below 350 K. Figure 2a shows the frequency response of the waveguide filter with a suppression of ~ -45 dB at 213.5 GHz."*

**Sequence number: 6**
**Author:**
**Date: 12/04/2019 14:15:57**
**do you mean "lower cost"?**
"cost" has been changed to "price" here.

**Sequence number: 7**
**Author:**
**Date: 12/04/2019 14:23:19**
**I would write the equation that relates the receiver noise to the system noise, to make clear the difference between the two parameters.**
Equations have been added to this section to clarify the difference between the receiver temperature and the system temperature. The radiometer equation is also included to relate the system temperature to the integration time.

*"The system temperature can be described as $T_{sys} = T_{rec} + T_a$ (Parrish et al., 1988, Janssen, 1993, Stanimirović et al., 2002). The receiver temperature, $T_{rec}$, considers the contributions from CORAM, and the antenna temperature, $T_a$, considers the contributions from the atmospheric background and signal being measured. The system temperature is related to the measurement time through the so-called radiometer equation: $\sigma_T = T_{sys} / (Bt)^{1/2}$, where $\sigma_T$ is the statistical noise on a measured spectrum, B is the frequency bandwidth of the measurement, and t is the integration time for the measurement."*

**Page: 4**
**Sequence number: 1**
**Author:**
**Date: 12/04/2019 14:40:49**
**I would add a short description of the observing equations (total power, correct?) and how the main unknowns in the equation are estimated/measured.**

Section 2 has been edited to include a description of the inversion problem and how it relates to the measurements made with CORAM:

*"2.2.1 Defining the inversion problem*

*Schwarzchild's equation describes radiative transfer through a medium in local thermodynamic equilibrium. In the millimetre-wave region, at a given frequency, the measured intensity can be expressed in terms of brightness temperature, $T_b$, where*

$$T_b = T_{b_0} e^{-\tau(l_0)} + \int_0^{l_0} T(l)\alpha(l)e^{-\tau(l)}dl, \tag{1}$$

*with l denoting the path through the atmosphere from a point $l_0$ to the measurement point at l = 0. The initial intensity is $T_{b_0}$, the optical depth of the atmosphere is described by $\tau$, and the absorption coefficient is defined as $\alpha$. More details can be found in Janssen (1993) and references therein. $T_b$ in equation (1), as a function of frequency, is generally the mathematical description of the calibrated atmospheric spectrum, the antenna temperature ($T_a$) from Sect. 2.1. For a total power radiometer such as CORAM, the calibrated antenna temperature is found using:*

$$T_a = \left(\frac{V_{atm}-V_c}{V_h-V_c}\right)(T_h - T_c) + T_c, \tag{2}$$

*where $T_h$ and $T_c$ are the temperatures of the hot and cold calibration targets (Sect. 2.1), $V_h$ and $V_c$ are the measured voltages when observing the hot and cold targets, respectively. $V_{atm}$ is the measured voltage when observing the atmosphere.*

*The desired quantity, the VMR of a trace gas, is contained within the description of the absorption coefficient, $\alpha$. Equation (1) must be inverted to retrieve this information. The form of Equation (1) is that of a Fredholm integral of the second kind and is inherently sensitive to small perturbations (like noise on a spectrum). To overcome this, the numerical inversion here is performed iteratively using a maximum a posterieri probability estimation.*

*2.2.2 Inversion method*

*Altitude profiles of CO VMR are retrieved from the measured spectra using an optimal estimation inversion technique (Rodgers, 2000). The method uses some a priori information of the state of the atmosphere to constrain the profile that is retrieved from the measured spectrum. The linear solution to the inversion problem can be expressed as*
$\hat{x} = Ax + (I - A)x_a$*, where $\hat{x}$ is the retrieved state vector (VMR profile), $x$ is the true atmospheric state vector, $x_a$ is the a priori state vector, and $I$ is the identity matrix. $A$ is the averaging kernel matrix, which describes the sensitivity of a retrieved state to the true state (Rodgers, 2000). The sensitivity of the retrieved state at altitude i, to the true state at altitude j, is given by $A_{ij} = \partial\hat{x}_i / \partial x_j$."*

**Sequence number: 2**
**Author:**
**Date: 12/04/2019 15:38:21**
**Given the large seasonal variability of mesospheric CO over polar regions, what will you do with summer data? Since you're**
**describing an instrument that is designed for long-term measurements you should plan for an entire year of data analysis.**
The CO concentrations during the summer are very low and are not detectable by CORAM. Clarification on this is now included in the abstract and in Section 2.1.

**Sequence number: 3**
**Author:**
**Date: 03/04/2019 13:02:25**
**This is true only if you consider the central part of the spectral line and not its broad wings which are produced from the emission of**
**stratospheric CO.**
A new line has been added directly after this to clarify that the broad wings of the spectral line are produced at altitudes lower than the retrievable altitude limit of CORAM.

*"The broad wings of a CO spectral line are produced by CO molecules at altitudes below the retrievable altitude limit of CORAM (approximately 47 km, see Sect. 2.3)."*

**Sequence number: 4**
**Author:**
**Date: 15/04/2019 16:14:11**

**it would be useful to see an example of the sinewaves that are being removed**
An example of the fit to the baseline is included in Figure 3.

**Page: 5**
**Sequence number: 1**
**Author:**
**Date: 03/04/2019 15:06:58**
**It's not clear whether the spectrum showed had already the sinewaves subtracted or not**
The spectrum shown is the original measurement. The fit to the baseline (baseline fit),
which includes the sinewaves, forms part of the inversion fit. The baseline fit is not
separately subtracted from the measurement.
The caption to Figure 3 has been edited to emphasise that the baseline fit in the lower panel
is a part of the overall fit shown in the upper panel.

From Section 2.2
*"Qpack2 provides the capability to fit a series of functions to the baseline of the measured
spectra (a baseline fit) to account for errors in the baseline which are likely caused by
standing waves in the instrument. The baseline fit is included in the optimal estimation and
forms part of the overall fit to the measurement (inversion fit)."*

*"Figure 3: (a) Upper: an example spectrum measured by CORAM on Dec 24th 2017 between
20:04 and 21:03 UTC. The inversion fit to the measurement is shown (smoother red line).
Lower: the residual of the measurement and the inversion fit (solid black line). The dashed
red line shows the baseline fit for the inversion, which is part of the inversion fit shown in the
upper panel (Sect. 2.2). (b) The CO profile retrieved from the measurement (solid blue) and
the a priori profile that is used as input to the inversion (dashed black)."*

**Sequence number: 2**
**Author:**
**Date: 15/04/2019 17:38:50**
**I think authors should be a little more precise here**
Section 2.2 has been edited to contain a more detailed description of the averaging kernels.

*"Altitude profiles of CO VMR are retrieved from the measured spectra using an optimal
estimation inversion technique (Rodgers, 2000). The method uses some a priori information*

*of the state of the atmosphere to constrain the profile that is retrieved from the measured spectrum. The linear solution to the inversion problem can be expressed as $\hat{x} = Ax + (I - A)x_a$, where $\hat{x}$ is the retrieved state vector, $x$ is the true atmospheric state vector, $x_a$ is the a priori state vector, and $I$ is the identity matrix. $A$ is the averaging kernel matrix, which describes the sensitivity of a retrieved state to the true state (Rodgers, 2000). The sensitivity of the retrieved state at altitude i, to the true state at altitude j, is given by $A_{ij} = \partial \hat{x}_i / \partial x_j$."*

**Sequence number: 3**
**Author:**
**Date: 15/04/2019 17:40:41**
**It's unclear to me how you can reach a 87 km altitude limit considering the Doppler broadening and with a 61 kHz channel resolution.**
The averaging kernels, which describe the distribution of sensitivity of the instrument are used. Section 2.3 outlines that a measurement response of 0.8 is often used to determine the altitude range of an instrument, as is done here. Later in Section 2.3, the peaks of the averaging kernels are discussed, and how this affects the interpretation of the CO profiles above ~70 km. This topic is discussed in more detail in Hoffmann et al. (2011) for ground-based CO measurements and this is also cited in Section 2.3.

*"A common way to estimate the altitude limits of a retrieved profile is to define the sum of the rows of the averaging kernels as the measurement response and assign a cut-off value. The choice of the cut-off value is rather arbitrary but 0.8 is regularly used (e.g., Forkmann et al., 2012; Straub et al., 2013, Schranz et al., 2018), and is also used here. With the above definitions, the CO profiles from CORAM during winter 2017/2018 have an average altitude range of approximately 47 – 87 km, with an average altitude resolution varying between approximately 12.5 and 28 km over that range. The retrieval range can change depending on the distribution of CO in the atmosphere (the lower limit can decrease in altitude when there are higher CO values at lower altitudes) and the value provided here is the mean range over the time span of the data.*
*The retrieval limits will vary from measurement to measurement and individual profiles should be considered in combination with the accompanying averaging kernels. The centres of the averaging kernels, when represented in VMR, are shifted down in altitude compared to a representation in relative units (Hoffmann et al., 2011). The lower limit of the retrieval here is defined by the SNR in the measurement and the upper limit is set by a transition from a pressure broadening regime to a doppler broadening one. The result of this change is that,*

*above approximately 70 km in the VMR representation, the centres of the averaging kernels do not increase in altitude with their corresponding retrieval altitudes. The retrieved CO values above ~ 70 km altitude do contain information from the atmosphere that corresponds with the retrieval altitude, but the VMR representation of the profile should be considered with care. Hoffmann et al. (2011) provides a detailed discussion on the representation of data for ground-based CO measurements. Hoffmann emphasises that the limited vertical resolution of the data must be taken into account for the use and interpretation of the data by considering each realisation of the averaging kernels, and so the a priori and averaging kernels form an essential part of the dataset."*

**Sequence number: 4**
**Author:**
**Date: 03/04/2019 15:26:27**
**Call figure 4**
This has been added.

**Sequence number: 5**
**Author:**
**Date: 15/04/2019 17:56:15**
**This sentence is unclear to me. As I see it, upwards of 70 km altitude you can't retrieve a profile anymore but have basically a**
**partial column content. This suggests that on top of using the measurement response between 0.8 and 1.2 in order to identify the**
**altitude range where the data sets is reliable, you should possibly use also the close correspondence between nominal and retrieval altitudes.**
The area under the averaging kernels (the sum of the rows) is used to define a limit. Then, later in Section 2.3, the correspondence between 'nominal and retrieval altitudes' is discussed. The location of peaks of the averaging kernels are discussed, and how this affects the interpretation of the CO profiles above ~70 km. This topic has been covered in more detail in Hoffmann et al. (2011) for ground-based CO measurements and this is also cited in Section 2.3.
"The retrieval limits will vary from measurement to measurement and individual profiles should be considered in combination with the accompanying averaging kernels. The centres of the averaging kernels, when represented in VMR, are shifted down in altitude compared to a representation in relative units (Hoffmann et al., 2011). The lower limit of the retrieval here is defined by the SNR in the measurement and the upper limit is set by a transition

from a pressure broadening regime to a doppler broadening one. The result of this change is that, above approximately 70 km in the VMR representation, the centres of the averaging kernels do not increase in altitude with their corresponding retrieval altitudes. The retrieved CO values above ~ 70 km altitude do contain information from the atmosphere that corresponds with the retrieval altitude, but the VMR representation of the profile should be considered with care. Hoffmann et al. (2011) provides a detailed discussion on the representation of data for ground-based CO measurements. *Hoffmann emphasises that the limited vertical resolution of the data must be taken into account for the use and interpretation of the data by considering each realisation of the averaging kernels, and so the a priori and averaging kernels form an essential part of the dataset."*

To make this clearer to the reader, the caveats on altitude range are now also included in both the abstract and the conclusion.

Abstract:
"The profiles in the current dataset have an average altitude range of 47-87 km, with special consideration to be given to data at > ~70 km altitude."

Conclusion:
*"The mean of the averaging kernel matrix for the CORAM dataset gives an average retrieval altitude range of 47-87 km with an average altitude resolution of 12.5 to 28 km over this range. Data at higher altitudes should be treated with care as the VMR representation of the averaging kernels do not peak at the corresponding retrieval grid points above ~70 km altitude."*

**Page: 6**
**Sequence number: 1**
**Author:**
**Date: 19/04/2019 18:02:11**
**This sentence suggests that the pointing azimuth of the instrument is unknown. Could you please clarify? Please explain how you**
**measure the elevation angle of the signal beam, how you set the zero elevation. Do you perform a sun scan? Authors write "The**
**atmospheric signal enters the lab at 20° elevation**
**and is directed by a series of mirrors through a window in a cryocooler". How do you measure the elevation angle above the**

**horizon of your beam?**
The sentence has been edited to clarify that the overestimate is used to account for changes that may occur in the orientation of the instrument table.
*"An uncertainty of 1° is chosen for the pointing of the instrument to the sky, an overestimate of the motor (Faulhaber 3564K024B CS) uncertainty by an order of magnitude, to account for changes that may occur in the orientation of the instrument table."*

Section 2.1 now includes information on the measurement of the elevation angle.
*"The alignment of the quasioptical components was checked using a laser positioned at the entrance to the cryocooler. The elevation angle of the instrument was measured using a self-levelling laser (Bosch GLL 3-80), which provides a horizontal line with an accuracy of 0.2 mm/m (0.2 mrad). Two horizontal lines, one directly from the laser and one passing through the quasioptical setup, were aligned on a screen approximately 5 m from the instrument. A sun scanning method has been used with other ground-based instruments to identify a pointing offset, e.g., for MIAWARA-C (Straub et al. 2010) and GROMOS-C (Fernandez et al., 2015), for which the offsets in the elevation angle were found to be 0.01° and 0.07°, respectively."*

**Sequence number: 2**
**Author:**
**Date: 19/04/2019 11:17:37**
**Are there temperature sensors measuring the various temps?**
This is now clarified in Section 2.1.

*"The measured signal is calibrated using two blackbody targets at known temperatures (measured with mounted sensors): a cold target in the cryocooler at ~ 70 K and a warm target at ~ 293 K."*

**Sequence number: 3**
**Author:**
**Date: 03/04/2019 15:25:30**
**Somewhere here there should be a call to Figure 5**
Figure 5 is called on line 13 of the original manuscript.
*"The error estimates, including the average of the error arising from statistical noise on the spectrum, are plotted in Fig. 5."*

**Page: 7**
**Sequence number: 1**
**Author:**
**Date: 19/04/2019 18:03:04**
**This smoothing process involves the CORAM apriori profile as well. For this reason, you cannot really calculate a correlation**
**coefficient between the MLS smoothed profiles and CORAM profiles as if the two datasets were independent. If you wish to do so, you should use the MLS original profiles or perform a smoothing process of MLS profiles which does not involve CORAM AVK or apriori.**
The correlation between the unsmoothed MLS data and CORAM data is now included in Figure 6. Section 3.1 has been edited to include the following.

*"After smoothing, the MLS and CORAM data are not truly independent, so the correlation of CORAM with the unsmoothed MLS data is also calculated and shows more variation over the retrievable altitude range, with a minimum of 0.59 and a maximum of 0.81."*

**Page: 9**
**Sequence number: 1**
**Author:**
**Date: 19/04/2019 12:45:25**
**I am uncomfortable with the overall statement that the valid altitude range for the retrieval is up to 87 km when it is well known that**
**above about 70 km altitude the Doppler broadening takes over and you cannot obtain a vertical distribution of CO from its line**
**shape.**
To clarify, the caveats on altitude range are now also included in both the abstract and the conclusion.

Abstract:
"The profiles in the current dataset have an average altitude range of 47-87 km, with special consideration to be given to data at > ~70 km altitude."

Conclusion:
*"The mean of the averaging kernel matrix for the CORAM dataset gives an average retrieval altitude range of 47-87 km with an average altitude resolution of 12.5 to 28 km over this*

*range. Data at higher altitudes should be treated with care as the VMR representation of the averaging kernels do not peak at the corresponding retrieval grid points above ~70 km altitude."*

**Sequence number: 2**
**Author:**
**Date: 19/04/2019 12:56:52**
**If you degrade MLS using CORAM averaging kernels and apriori the two datasets are then not independent and you can't really**
**talk about their "correlation". See earlier comment.**
The statement has been edited to include information on the smoothed and unsmoothed data.

*"Correlations between the instruments range from 0.80 to 0.92 over CORAMs retrievable altitude range for MLS data smoothed with the CORAM averaging kernels, and from 0.59 to 0.81 when using the unsmoothed MLS data."*

**Page: 15**
**Sequence number: 1**
**Author:**
**Date: 03/04/2019 12:23:03**
**Remove "in"**
This has been fixed.

**Sequence number: 2**
**Author:**
**Date: 12/04/2019 14:23:52**
**why was this measurement carried out at 8.5 GHz and not at the FFTS?**
The measurement was made by RPG as part of the production process for the new components. The system temperature of CORAM of ~600 K is now also included in the caption.

**Page: 16**
**Sequence number: 1**
**Author:**
**Date: 03/04/2019 13:07:32**

**I do not understand whether the dashed red line represents what was subtracted from the original spectrum. Authors should explain/show this subtraction a little better as this is always a touchy topic.**

The dashed red line is the fit to the baseline that is included in the inversion fit (the overall fit of the line). A separate subtraction is not made and is purposefully not mentioned in the description in Section 2.2 nor in the caption to Figure 3. It is now emphasized in the caption that the baseline fit is a part of the inversion fit shown in the upper panel.

From Section 2.2

*"Qpack2 provides the capability to fit a series of functions to the baseline of the measured spectra (a baseline fit) to account for errors in the baseline which are likely caused by standing waves in the instrument. The baseline fit is included in the optimal estimation and forms part of the overall fit to the measurement (inversion fit)."*

*"Figure 3: (a) Upper: an example spectrum measured by CORAM on Dec 24th 2017 between 20:04 and 21:03 UTC. The inversion fit to the measurement is shown (smoother red line). Lower: the residual of the measurement and the inversion fit (solid black line). The dashed red line shows the baseline fit for the inversion, which is part of the inversion fit shown in the upper panel (Sect. 2.2). (b) The CO profile retrieved from the measurement (solid blue) and the a priori profile that is used as input to the inversion (dashed black)."*

**Page: 17**
**Sequence number: 1**
**Author:**
**Date: 15/04/2019 16:45:24**
**In my understanding measurement response values larger than 1.2 are as critical as those below 0.8. Is this correct?**

The measurement response can be thought of as a rough measure of the fraction of the retrieved state that comes from the data, instead of from the a priori. That is why it is often used to determine a cutoff where the data contribution is considered too little. It is only a rough measure though, as seen, and as you pointed out, by the measurement response often exceeding 1 at some altitudes.

More information has been added to Section 2.3 and reference to Rodgers (2000) and Payne et al. (2009).

*"The measurement response can generally be thought of as a rough measure of the fraction of the retrieved state that comes from the data, rather than the a priori (Rodgers., 2000). As noted by Payne et al. (2009), this is only a rough measure, and the measurement response often exceeds 1 at some altitudes."*

**Page: 18**
**Sequence number: 1**
**Author:**
**Date: 19/04/2019 18:04:53**
**I am very surprised that a pointing uncertainty of 1° leads to such a small uncertainty in the retrieved profile.**
The calculations were checked and the same result was found. It is likely that the pointing is more critical for systems that use a tipping curve method to calculate the atmospheric opacity for use in correcting the measured spectrum. And also for systems that use atmospheric measurements at one or more specific angles as the hot/cold targets to calibrate the signal data.

**Page: 20**
**Sequence number: 1**
**Author:**
**Date: 19/04/2019 18:19:27**
**I think authors should show these time series at various altitudes so that the reader can better evaluate the difference between**
**CORAM and MLS datapoints. The vertical scales here are so different from altitude to altitude that it is really difficult to grasp useful**
**info.**
It is unclear what is meant here. The time series is shown at 5 altitudes between 48 and 88 km. To clarify, the figure caption has been expanded to include the specific altitudes.

*"Figure 7: Time series of the daily CORAM and MLS CO VMR values at altitudes of 48, 58, 68, 78, and 88 km."*

---

## Author Response (AR2)

**Response to second comments from Referee #2**

**PG 2, LN 15-17: There are molecules being repeated. The wording is awkward.**
This line, from the original manuscript, provides some examples of projects using tracers to observe downward transport at the poles. To identify the tracers used in each project, there is a choice between using some citations more than once or naming some gases more than once. We have chosen the latter.

**PG 4, LN 5-8: This technique for estimating the pointing offset has, however, nothing to do with CORAM. I don't see the point of mentioning it.**
The original comments from Reviewer #2 showed much concern over the method used to determine the absolute elevation angle of the measurement. The method used in this work makes use of a levelling instrument. For the interest of readers who might not have access to this type of instrument, we also include citations to works that perform this calibration using a sun scanning method.
The sentence has been edited to the following, to emphasize that the cited technique is used for alignment:
*"A sun scanning method has been used with other ground-based instruments for alignment, and to identify a pointing offset, e.g., for MIAWARA-C (Straub et al. 2010) and GROMOS-C (Fernandez et al., 2015), for which the offsets in the elevation angle were found to be 0.01°and 0.07°, respectively."*

**PG 9, LN 5-9: This comparison results should be corrected with numbers showed by Sheese et al., JQSRT, 2017. In particular, just by reading their abstract you learn that "With respect to winter MLS CO, ACE-FTS is typically within +/- 10% between 25 and 40 km, and has an average bias of -11% above 40 km."**
The description has been edited to read:
*"The data have an estimated average positive bias (larger VMRs) of ~17% above 40 km altitude (Sheese et al., 2017), compared to the Atmospheric Chemistry Experiment – Fourier Transform Spectrometer (ACE-FTS) satellite instrument. Sheese et al. (2017) use Version 3.3/3.4 MLS CO data, which shows good agreement with Version 4.2 (Livesey et al., 2018), and have not included data from the summer months when CO concentrations are very low."*

**Authors should check the list of citations as there seem to be missing ones.**
The missing references for the citations have been added.

[revised manuscript text omitted]